# *SAFE*: Multitask Failure Detection for Vision-Language-Action Models

**Qiao Gu**[1,2,3]   **Yuanliang Ju**[1,2,3]   **Shengxiang Sun**[1,2]   **Igor Gilitschenski**[1,2,3]
**Haruki Nishimura**[4]   **Masha Itkina**[4]   **Florian Shkurti**[1,2,3]
[1]University of Toronto (UofT), [2]UofT Robotics Institute,
[3]Vector Institute, [4]Toyota Research Institute (TRI)
`q.gu@mail.utoronto.ca`

## Abstract

While vision-language-action models (VLAs) have shown promising robotic be-haviors across a diverse set of manipulation tasks, they achieve limited success rates when deployed on novel tasks out of the box. To allow these policies to safely interact with their environments, we need a failure detector that gives a timely alert such that the robot can stop, backtrack, or ask for help. However, existing failure detectors are trained and tested only on one or a few specific tasks, while generalist VLAs require the detector to generalize and detect failures also in unseen tasks and novel environments. In this paper, we introduce the multitask failure detection problem and propose *SAFE*, a failure detector for generalist robot policies such as VLAs. We analyze the VLA feature space and find that VLAs have sufficient high-level knowledge about task success and failure, which is generic across different tasks. Based on this insight, we design *SAFE* to learn from VLA internal features and predict a single scalar indicating the likelihood of task failure. *SAFE* is trained on both successful and failed rollouts, and is evaluated on unseen tasks. *SAFE* is compatible with different policy architectures. We test it on OpenVLA, $\pi_0$, and $\pi_0$-FAST in both simulated and real-world environments extensively. We compare *SAFE* with diverse baselines and show that *SAFE* achieves state-of-the-art failure detection performance and a favorable trade-off between accuracy and detection time using conformal prediction. More qualitative results and code can be found at the project webpage: https://vla-safe.github.io/.

## 1   Introduction

Recently, scaling up robot manipulation datasets has enabled the development of large vision-language-action (VLA) models, which are generalist manipulation policies that can follow language instructions and accomplish a wide range of tasks [1–6]. However, when VLAs are directly deployed on unseen tasks without collecting additional demonstrations and finetuning the model, they still suffer from limited success rates and a wide range of failure modes. This has been demonstrated by evaluations in recent work [2, 4, 7]: while VLAs achieve success rates of 80–90% on seen tasks, their performance on unseen tasks drops to 30–60% out of the box. Therefore, to safely and reliably deploy VLA policies in the real world, it is important to promptly detect their potential failures.

Most existing failure detection methods train a separate failure detector for each task, and evaluate the detector only on that task [8–17]. While these methods work well for specialist policies, they do not suit generalists like VLAs. VLAs are designed to accomplish diverse tasks and may frequently encounter novel task instructions and unseen environments during deployment. In such cases, it is impractical to exhaustively collect rollouts and train a failure detector for every new task. Some recent works introduce task-generic failure detectors, but they either require sampling multiple actions [18]

or need to query a large VLM [19, 20], which poses significant inference overhead for VLAs in the real world. This motivates the need for an *efficient* and *multitask* failure detector that can generalize to unseen tasks zero-shot and detect failures in a timely manner during the on-policy rollout of the VLA.

In this paper, we focus on the *multitask* failure detection problem. This setting evaluates the failure detection performance of a VLA policy without collecting rollouts or finetuning the failure detector on unseen tasks. To our knowledge, such multitask failure detection capabilities for VLAs have not been shown in the literature. To tackle this problem, we study the internal features of VLAs and find that they capture high-level knowledge about task success and failure. As shown in Fig. 1, failed rollouts occupy a distinct region ("failure zone") in the VLA feature space, and this separation remains consistent across different tasks.

Based on this insight, we introduce *SAFE*, a ScAlable Failure Estimation method that scales across diverse tasks for generalist policies like VLAs. *SAFE* takes in a VLA's internal features and regresses them to a single scalar indicating the likelihood of failure. By training on successful and failed rollouts of multiple tasks, *SAFE* learns to identify task-generic representations for failure detection. To determine the threshold for failure detection, we adopt the functional conformal prediction (CP) [8, 21] framework and calibrate the prediction band on the seen tasks. We conduct failure detection experiments on OpenVLA [2], $\pi_0$ [4] and $\pi_0$-FAST [5], in both simulation and the real world. For evaluation, we adapt diverse baseline failure detection methods from both the LLM literature [22, 23] and the robot learning literature [8, 18] onto VLAs. *SAFE* and baselines are evaluated on both training (Seen) tasks and a set of held-out (Unseen) tasks. Experiments show that *SAFE* outperforms other existing baselines and achieves the best trade-off between accuracy and timeliness for failure detection. The contributions of our paper can be summarized as follows:

- We analyze the VLA feature space and show that, across different task instructions and environments, the internal features of the VLA distinctly separate successful and failed rollouts.

- We propose *SAFE*, a multitask failure detector designed for generalist robot policies. By operating on latent features, training on multiple tasks, and using conformal prediction methods, *SAFE* shows generalization capabilities in detecting failures on unseen tasks.

- We evaluate *SAFE* and diverse baselines on several recent large VLA models in both simulation and the real world. Experiments show that *SAFE* outperforms baselines and achieves state-of-the-art (SOTA) performance.

## 2 Related Work

### 2.1 Vision-Language-Action Models

Recent advances in large-scale machine learning and the availability of extensive robot demonstration datasets have paved the way for VLA models [1–4, 6, 7, 24, 25]. These generalist robotic policies are initialized from pretrained large-scale VLMs [26–28], and thus inherit the ability to understand diverse semantic concepts from both images and language. They are augmented with an action head that produces continuous control signals, through per-step binning [1, 2, 7, 24], diffusion networks [3, 4, 29, 30], or frequency-space tokenization [5]. These VLAs are then trained on vast robotic datasets covering a wide array of tasks [31–33]. As a result, VLAs can successfully perform familiar tasks in new environments and even tackle previously unseen tasks when provided with novel language instructions. Nevertheless, significant variability in real-world deployments and the challenging domain gaps between training and testing environments continue to hinder VLA performance. Most state-of-the-art VLA models achieve success rates between 30% and 60% when evaluated out-of-the-box on real robots with unseen task instructions [2, 4, 31]. These limitations highlight the need for robust multitask failure detection methods tailored to generalist VLA models.

### 2.2 Failure Detection in Robot Manipulation

Monitoring failures is critical when deploying robotic policies in real-world environments, as even minor errors can result in hazardous conditions [34–36]. The literature on failure detection in robot learning can be broadly divided into unsupervised out-of-distribution (OOD) detection [8–11, 17] and

supervised failure detection [9, 12–16, 37]. OOD detection-based methods treat successful executions as the in-domain baseline and consider any deviation from this norm as a failure. However, the assumption that any unseen scenario constitutes a failure is overly restrictive for generalist VLAs, which may frequently encounter unseen tasks at test time. These unseen tasks are likely different from the in-domain training data but should not be simply treated as failures. Our proposed method, *SAFE*, falls within the supervised failure detection category, leveraging both successful and failed rollouts to train a failure classifier. Unlike existing methods that train and calibrate separate classifiers per task, *SAFE* uses a single unified failure detector and works effectively on generalist policies like VLAs. Some recent works have explored multitask failure detection by designing action consistency scores [18] or instruction-finetuning a VLM [19, 20], but they require either sampling multiple actions or querying a large VLM, which poses significant overhead for controlling robots in real time.

Recently, FAIL-Detect [8] conducted a systematic evaluation of various failure detection methods, including OOD detection-based approaches [9, 38, 39], smoothness-based techniques [40], and consistency-based strategies [18]. Their experiments indicate that the best performance was achieved by LogpZO, which learns a proxy for the likelihood of the data in the observation embedding space using flow matching [8]. However, their evaluation is limited to only single-task policies, and our evaluation in the multitask setting shows that their best-performing LogpZO method suffers from overfitting to the training tasks.

### 2.3 Uncertainty Quantification for LLM

Although LLMs and VLMs have demonstrated remarkable understanding and generative capabilities across various tasks, they are prone to producing hallucinated responses [41–43]. Numerous methods have been developed for uncertainty quantification (UQ) in LLMs/VLMs. Token-level uncertainty quantification methods estimate uncertainty by analyzing the probability distribution over each generated token to assess the likelihood of an entire response [44–46]. Semantic-similarity methods generate multiple responses to the same query and evaluate their semantic alignment [23, 47, 48]; a higher variance among responses typically signals low confidence. Since vision-language-action models (VLAs) share the generative nature and transformer architecture of LLMs/VLMs, we adapt these UQ methods to VLAs as promising baselines and evaluate their performance on failure detection. Note that these UQ baselines are used as a proxy for failure detection, assuming that when a policy becomes uncertain about its actions, it will have a higher probability of failing the task. Recent research has also explored the internal latent space of LLMs for hallucination detection [49–55]. These methods train a classifier on internal latent features to distinguish between truthful and hallucinated outputs, paralleling supervised failure detection techniques in robotics. This approach has proven to be simple, efficient, and effective for UQ in LLMs. In our study, we investigate its application to large VLA policies and observe promising performance in robotic tasks.

## 3 Problem Formulation

This work aims to detect when a robot policy fails during task execution. Specifically, we develop a multi-task failure detector that performs well when generalist VLAs encounter novel tasks at inference time. At timestep $t$, a VLA is given an input observation $\mathbf{o}_t$, consisting of RGB images, natural language instruction, and current robot state, and outputs a control signal $\mathbf{A}_t = [\mathbf{a}_t, \mathbf{a}_{t+1}, \ldots, \mathbf{a}_{t+H-1}]$, which is a chunk of actions for the next $H$ timesteps. The first $H'$ ($H' \leq H$) actions in $\mathbf{A}_t$ are executed, and then the VLA replans a new action sequence $\mathbf{A}_{t+H'}$ at time $t + H'$. We denote the internal embedding vector within the VLA model at time $t$ as $\mathbf{e}_t$. Some VLAs [1, 2, 4, 5] also decode a series of $m$ tokens $\mathbf{W}_t = [\mathbf{w}_t^1, \ldots, \mathbf{w}_t^m]$ before converting them into the actual action vector. To train and evaluate failure detection models, we run the VLA on different tasks in simulation or the real world, collect the rollout trajectory $\tau_i = \{(\mathbf{o}_t, \mathbf{e}_t, \mathbf{W}_t, \mathbf{A}_t)\}_{t=0, H', \ldots, nH'}$ with time duration $T = nH'$, and annotate each rollout with a failure label $y_i$ ($y_i = 1$ if the robot fails to accomplish the task and $y_i = 0$ if the robot succeeds). Note that for training, we only use the trajectory-level annotation $y_i$, and do not require knowing the exact timestep when the policy starts to fail. A failure detector receives the rollout information up to time $t$ and predicts a failure score $s_t$, indicating the likelihood of task execution failure at time $t$. If $s_t$ exceeds a threshold $\delta_t$, a failure flag is raised, and then either the task execution is aborted or a human monitor will step in and take over the control. In this work, we use conformal prediction [58] to calibrate the threshold $\delta_t$.

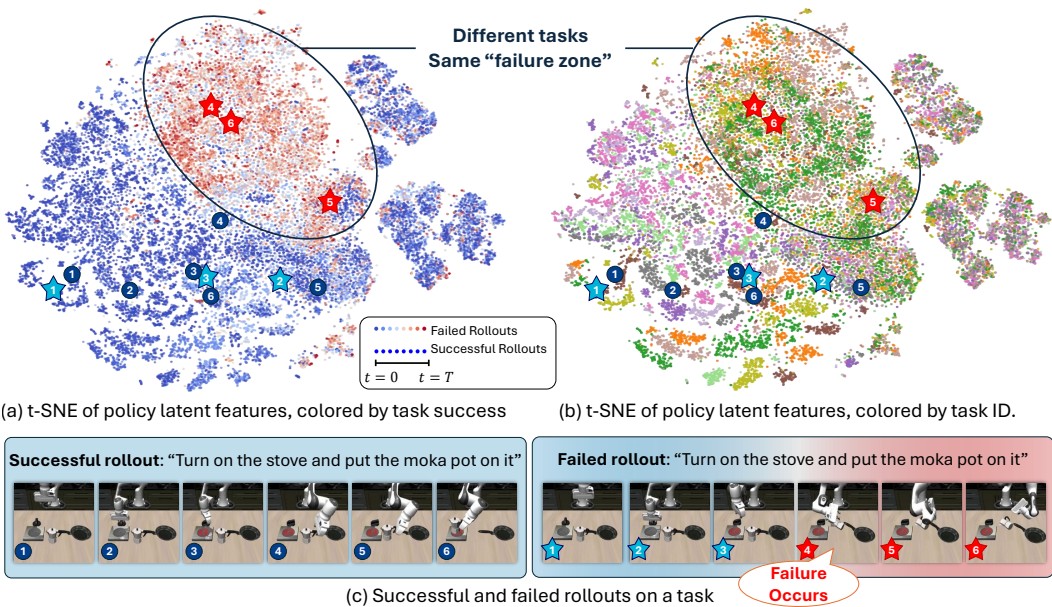

(a) t-SNE of policy latent features, colored by task success

(b) t-SNE of policy latent features, colored by task ID.

(c) Successful and failed rollouts on a task

Figure 1: The internal features of a VLA capture high-level information about task success and failure. When the VLA is failing, the features, even those from different tasks, fall into the same "failure zone". This motivates *SAFE*, an efficient multitask failure detector that is based on VLA internal features and can generalize to unseen tasks. Plot (a) visualizes the latent features of $\pi_0$-FAST on LIBERO-10 [56] using t-SNE [57]. For successful rollouts, features are colored in blue. For failed rollouts, features follow a blue-to-red gradient based on timestep progression, with red corresponding to later timesteps that often coincide with failure. Plot (b) visualizes the same set of t-SNE features, colored by task ID. In (c), we show two example rollouts over time and mark their corresponding projected features in (a) and (b).

In experiments, we split all tasks into *seen* and *unseen* subsets, where rollouts from seen tasks are used for training $\mathcal{D}_{\text{train}}$ and validation $\mathcal{D}_{\text{eval-seen}}$, and all rollouts from unseen tasks $\mathcal{D}_{\text{eval-unseen}}$ are reserved for testing the cross-task generalization ability of failure detectors. Failure detectors are trained on $\mathcal{D}_{\text{train}}$, and evaluated on $\mathcal{D}_{\text{eval-seen}}$ for hyperparameter tuning and in-domain performance, and tested on $\mathcal{D}_{\text{eval-unseen}}$ for out-of-distribution generalization.

# 4 Method

## 4.1 Visual Analysis on VLA Latent Space

VLAs process multi-modal inputs and extract rich semantic information in their internal feature space. We hypothesize that these features also capture the high-level and abstract knowledge about task execution success/failure, by separating features from successful/failed rollouts into different regions. We study this hypothesis by visualizing the VLA features in Fig. 1, where we plot the internal features from $\pi_0$-FAST[5] when running the LIBERO-10 benchmark [56]. Fig. 1(a) demonstrates that when the VLA is failing, its internal features are grouped in the same region in the feature space ("failure zone"). Comparing Fig. 1(a) and Fig. 1(b), we can further see that although the features are extracted from different tasks with various instructions, objects and environments, when the VLA fails, its features fall in the same "failure zone". Fig. 1(c) further illustrates how VLA's features evolve in the feature space when VLA progresses temporally. From Fig. 1(c), we can see that failure rollout initially stays out of the "failure zone" when it progresses normally, and when the robot mistakenly drops the pot in the middle of execution and starts to fail, it steps into the "failure zone". On the contrary, for the successful rollout, its features always stay out of the "failure zone".

This visual analysis shows that the VLA's internal features for succeeding and failing task executions are well separated in the feature space, and this separation is general across different tasks. Furthermore, during task execution, the features reflect how well the VLA performs on the current tasks in a

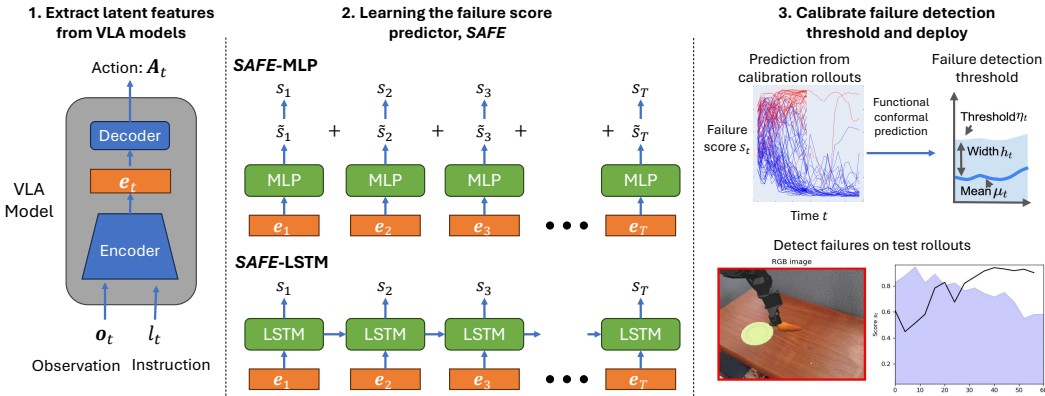

Figure 2: The proposed failure detector, *SAFE*, has three major components: (1) *SAFE* extracts the latent feature from the last layer of a VLA model; (2) *SAFE* sequentially processes the latent feature and predicts a failure score, using an MLP or an LSTM backbone; and (3) SAFE determines a time-varying threshold using functional conformal prediction (CP) on a hold-out calibration set. If the predicted score exceeds the threshold during testing, SAFE confidently detects a failure.

timely manner. Inspired by this observation, we design *SAFE*, which uses the internal features of VLAs for failure detection. An overview of the proposed method is shown in Fig. 2.

## 4.2 Failure Detection by Feature Probing

We design *SAFE* to learn the abstract information from the VLA's internal features and determine whether the task execution is failing. We extract the VLA's hidden state vectors from the final layer, before being decoded to token logits [2, 5] or a velocity field [4]. We ablate different ways to aggregate the internal features into a single embedding vector $\mathbf{e}$, and select the best one based on $\mathcal{D}_{\text{eval-seen}}$ performance. Please refer to Appendix for details on VLA feature extraction.

The failure detector $f(\mathbf{e}_{0:t})$ takes as input the VLA's features $\mathbf{e}_{0:t} = \{\mathbf{e}_1, \ldots, \mathbf{e}_t\}$ up to the current timestep $t$, and is trained to predict $s_t$. We explore the two backbone designs for *SAFE*: a multi-layer perceptron ($f_{\text{MLP}}$) and an LSTM [59] ($f_{\text{LSTM}}$). Both models are designed to be simple (only one or two layers), to avoid overfitting and improve generalization ability on unseen tasks. For $f_{\text{MLP}}$, we use an MLP $g(\cdot)$ to project $\mathbf{e}_t$ into a single scalar for each timestep $t$ independently and accumulate the outputs as the failure score, i.e. $f_{\text{MLP}}(\mathbf{e}_{0:t}) = \sum_{\tau=1,\ldots,t} \sigma(g(\mathbf{e}_\tau))$, where $\sigma(\cdot)$ is a sigmoid function and therefore $0 < s_t < t$. To train the MLP model, we apply an L1 loss on all timesteps to push up the scores for failed rollouts and push down those for successful ones. Specifically, $L_{\text{MLP}} = \sum_i \left[ y_i \sum_t (t - s_t) + (1 - y_i) \sum_t s_t \right]$, where index $i$ iterates over all data points in $\mathcal{D}_{\text{train}}$.

For $f_{\text{LSTM}}$, we use an LSTM model to sequentially process the input stream of VLA's features $\mathbf{e}_{0:t}$ and project the hidden state vector of LSTM into a scalar score. Specifically, $f_{\text{LSTM}}(\mathbf{e}_{0:t}) = \sigma(\text{LSTM}(\mathbf{e}_{0:t}))$, where a sigmoid function $\sigma(\cdot)$ is applied to normalize the output score s.t. $0 \leq s_t \leq 1$. To train the LSTM model, we apply a binary cross entropy loss on all timesteps, i.e. $L_{\text{LSTM}} = \sum_i \sum_t \left[ y_i \log(s_t) + (1 - y_i) \log(1 - s_t) \right]$.

## 4.3 Threshold Selection by Conformal Prediction

When the predicted failure score $s_t$ exceeds the time-varying threshold $\delta_t$, we raise a failure flag. To determine $\delta_t$ in a principled way, we adopt the functional conformal prediction (CP) framework [21]. Functional CP constructs a time-varying prediction band by leveraging the distribution of $s_t$ observed in successful rollouts within a calibration set. Under the exchangeability assumption [60] and given a user-specified significance level $\alpha$, the CP band guarantees that, for a new successful rollout, its $s_t$ will lie within this band at all times $t$ with probability $1 - \alpha$. Conversely, if the score of a test rollout exceeds the band at time $t$, we can declare a failure with nominal confidence $1 - \alpha$.

Formally, given a time series of any scalar score $s_t$ and a user-specified significance level $\alpha \in (0, 1)$, functional CP gives a distribution-free prediction band $C_\alpha$. Following Xu et al. [8], we adopt the one-sided time-varying CP band formulation, where $C_\alpha$ is a set of intervals $\{[\text{lower}_t, \text{upper}_t] : t = $

Table 1: Failure detection results on simulation benchmarks, measured by area under ROC (ROC-AUC). "-" indicates that the failure detection method does not apply. Entries with gray background indicate the failure detection methods that sample 10 actions per inference timestep, while others use only 1 action. The **first** and second best-performing methods are colored in red and orange, respectively. Results are averaged over 3 random seeds with different splits of seen and unseen tasks. *SAFE* achieves the highest averaged ROC-AUC over all simulation benchmarks.

| | VLA Model | OpenVLA | | $\pi_0$-FAST | | $\pi_0$ | | $\pi_0^*$ | | Average | |
| | Benchmark | LIBERO | | LIBERO | | LIBERO | | SimplerEnv | | | |
| | Eval Task Split | Seen | Unseen | Seen | Unseen | Seen | Unseen | Seen | Unseen | Seen | Unseen |
|---|---|---|---|---|---|---|---|---|---|---|---|
| Token Unc. | Max prob. | 50.25 | 53.83 | 61.32 | 69.44 | - | - | - | - | 55.79 | 61.64 |
| | Avg prob. | 44.05 | 51.58 | 52.46 | 58.04 | - | - | - | - | 48.26 | 54.81 |
| | Max entropy | 52.94 | 53.09 | 46.69 | 62.96 | - | - | - | - | 49.81 | 58.03 |
| | Avg entropy | 45.27 | 50.03 | 50.93 | 58.63 | - | - | - | - | 48.10 | 54.33 |
| Embed. Distr. | Mahalanobis dist. | 62.03 | 58.85 | **93.56** | 83.79 | **77.12** | **74.31** | 88.42 | 52.84 | 80.28 | 67.45 |
| | Euclidean dist. $k$-NN | 66.00 | 55.23 | 92.04 | 84.12 | 75.64 | 70.73 | 89.73 | 68.41 | 80.85 | 69.62 |
| | Cosine dist. $k$-NN | 67.09 | 69.45 | 92.09 | 84.64 | 75.76 | 70.31 | 90.19 | 71.32 | 81.28 | 73.93 |
| | PCA-KMeans [9] | 57.18 | 55.10 | 68.46 | 57.12 | 64.92 | 60.35 | 66.88 | 61.19 | 64.36 | 58.44 |
| | RND [39] | 52.57 | 46.88 | 88.67 | 81.57 | 71.92 | 69.44 | 85.07 | 65.89 | 74.56 | 65.95 |
| | LogpZO [8] | 61.57 | 52.91 | 91.52 | 83.07 | 76.80 | 73.23 | 88.79 | 74.66 | 79.67 | 70.97 |
| Sample Consist. | Action total var. | 62.76 | 65.43 | 76.95 | 74.50 | 77.20 | 75.18 | 68.41 | 67.94 | 71.33 | 70.76 |
| | Trans. total var. | 55.33 | 58.99 | 78.21 | 80.03 | 49.38 | 54.71 | 63.27 | 55.90 | 61.55 | 62.41 |
| | Rot. total var. | 47.85 | 55.30 | 80.87 | 77.29 | 52.94 | 61.06 | 58.07 | 62.10 | 59.93 | 63.94 |
| | Gripper total var. | 61.84 | 64.48 | 76.82 | 74.42 | 77.19 | 75.19 | 69.16 | 69.29 | 71.25 | 70.84 |
| | Cluster entropy | 50.16 | 51.44 | 80.22 | 80.53 | 76.19 | 72.12 | 68.25 | 73.66 | 68.71 | 69.44 |
| Action Consist. | STAC [18] | - | - | 83.07 | 85.31 | 46.55 | 47.91 | 60.74 | 62.21 | 63.45 | 65.14 |
| | STAC-Single | - | - | 85.46 | 81.16 | 68.46 | 69.39 | 68.71 | 70.40 | 74.21 | 73.65 |
| *SAFE* (Ours) | *SAFE-LSTM* | 70.24 | 72.47 | 92.98 | 84.48 | 76.98 | 71.09 | 88.85 | 80.11 | 82.26 | 77.04 |
| | *SAFE-MLP* | 72.68 | 73.47 | 90.06 | 80.44 | 73.50 | 73.27 | 89.50 | 84.82 | 81.43 | 78.00 |

$1, \ldots, T\}$, where $\text{lower}_t = -\infty$ and $\text{upper}_t = \mu_t + h_t$, with a time-varying mean $\mu_t$ and a bandwidth $h_t$. This band is calibrated on successful rollouts in $\mathcal{D}_{\text{eval-seen}}$. Under mild assumptions [61, 62], for any new successful rollout, $s_t < \mu_t + h_t$ holds for all $t = 1, \ldots, T$ with probability $1 - \alpha$. Intuitively, this gives a guarantee that the false positive rate of the failure detector (a failure flag is raised at any time during a successful rollout) is at most $\alpha$. We use $\text{upper}_t$ as the failure flag threshold $\delta_t$, and more details about functional CP can be found in Appendix.

## 5 Experiments

### 5.1 Evaluation Benchmarks

**LIBERO** [56]: The LIBERO benchmark has been widely adopted for evaluating VLA models in simulation [2, 4–6]. Among the LIBERO task suites, the LIBERO-10 suite consists of 10 long-horizon tasks with diverse objects, layouts, and instructions, and is considered the most challenging one. Therefore, we use LIBERO-10 in our experiments and test OpenVLA [2], $\pi_0$ [4] and $\pi_0$-FAST [5] on it. We adopt the model checkpoints that are finetuned on the LIBERO benchmark and released by their authors. In experiments, 3 out of 10 tasks are randomly picked and reserved as unseen tasks.

**SimplerEnv** [63]: SimplerEnv provides a high-fidelity simulation environment for manipulation policies, which are replicas of the demonstration data from RT-series [1, 7, 31] and BridgeData V2 [33]. On SimplerEnv, we test pretrained $\pi_0$ models from a reproduction [64], which we denote as $\pi_0^*$ in this paper. We train and evaluate the failure detection methods on the Google Robot embodiment [1] and on the WidowX embodiment [33], respectively. We exclude the "pick up coke" task because $\pi_0^*$ rarely fails on it (success rate at 98%). This leaves 4 tasks for each embodiment, among which 3 tasks are seen and 1 task is unseen.

**Real-world Franka Experiments**: We deploy the $\pi_0$-FAST-DROID checkpoint [4, 5][1] on a Franka Emika Panda Robot. This checkpoint has been finetuned on the DROID dataset [32], and we do not further collect demonstrations or finetune the VLA model. We design 13 tasks and collect 30 successful and 30 failed rollouts for each task. The real-robot setup and example rollouts are visualized in Fig. 3. In experiments, 3 tasks out of 13 are randomly selected as unseen tasks.

**Real-world WidowX Experiments**: We also deploy the OpenVLA model pretrained on the "Open-X Magic Soup++" dataset [2] on a WidowX robot manipulator in our lab. With this setup, we collected

---

[1]https://github.com/Physical-Intelligence/openpi

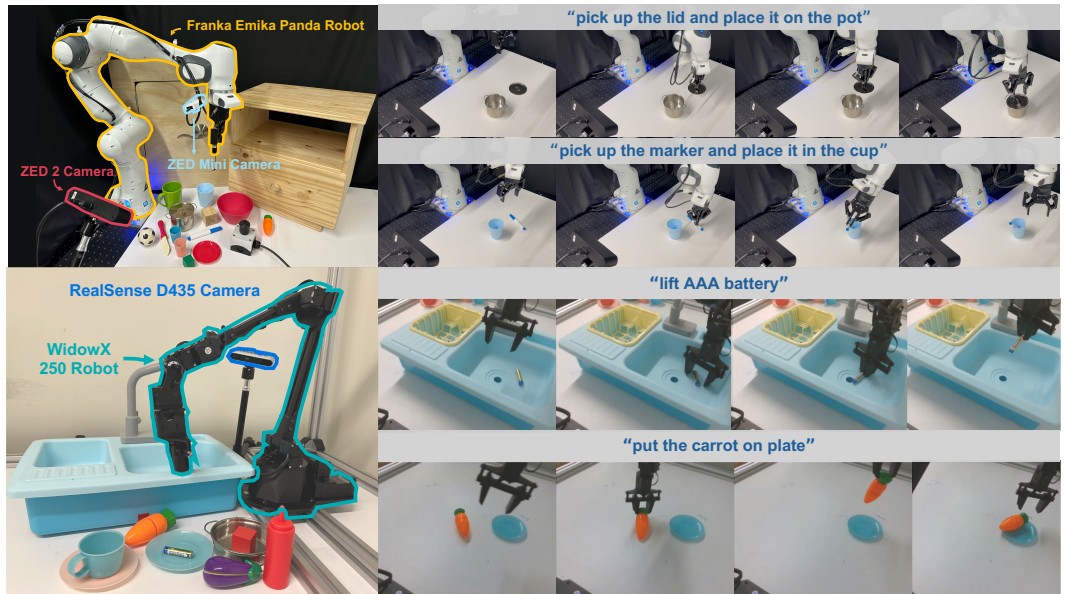

Figure 3: Illustration of real-world experiment setup (left) and example rollouts collected (right).

a total of 532 rollouts on the 8 lifting and pick-and-place tasks, including 244 successful and 288 failed rollouts. In this experiment, 2 tasks out of 8 are randomly selected as unseen tasks.

## 5.2 Uncertainty Quantification Baselines

Estimating uncertainty in generated responses has been widely used to detect truthfulness or hallucination in LLMs [22, 23, 43, 65]. For VLAs, uncertainty in the generated actions may indicate a lack of ability to solve the given task, and thus correlates with task failures. Therefore, we first adapt the UQ methods from the LLM literature to VLAs and use them as failure detection baselines.

**Token uncertainty**-based methods aggregate the predictive uncertainty from each generated token. These methods are efficient, as they only require a single forward inference. Given the generated tokens $\mathbf{W}_t = [\mathbf{w}_t^1, \dots, \mathbf{w}_t^m]$, we denote the probability of sampling the token $\mathbf{w}_t^i$ as $p_i$ and the entropy over the distribution of the $i^{\text{th}}$ token as $H_i$. We adopt the token-based uncertainty estimation methods used by Huang et al. [22] as follows:

$$\text{Token max prob.: } \max_i(-\log p_i); \quad \text{Token avg prob.: } -\frac{1}{m}\sum_i \log p_i;$$
$$\text{Token max entropy: } \max_i H_i; \quad \text{Token avg entropy: } \frac{1}{m}\sum_i H_i.$$

**Sample consistency**-based methods estimate uncertainty as the inconsistency within multiple generated sentences [22, 23, 65]. For VLA models, the output actions are continuous vectors, and we can measure inconsistency by their variance. Specifically, at time $t$, given $K$ sampled actions $\mathcal{A}_t = \{\mathbf{A}_t^k\}_{k=1,\dots,K}$, we measure the uncertainty as the total variation over the set of vectors: *action total var.* $= \text{trace}(\text{cov}(\mathcal{A}_t))$. Similarly, we also compute variation for the translational (*trans. total var.*), rotational (*rot. total var.*), and gripper control (*gripper total var.*) components of $\mathcal{A}_t$.

Furthermore, inspired by semantic entropy [23], we define *cluster entropy* as entropy(cluster($\{\mathbf{A}_t^k\}_{k=1,\dots,K}$)), where cluster($\cdot$) generates an integer set, containing $k$ cluster labels for the $k$ actions and entropy($\cdot$) measures the entropy of the integer set. UQ methods based on sample consistency are shown to perform better for LLM [22, 23], but they necessitate multiple inferences, which may not be practical for large VLAs that control robots in real time.

## 5.3 Failure Detection Baselines

**Embedding Distance**: We compare to baselines that directly use the distances in the feature space as failure scores. Specifically, instead of training a neural network, all VLA embeddings from $\mathcal{D}_{\text{train}}$ are stored in the two feature sets, $E_{\text{succ}}$ and $E_{\text{fail}}$, containing all VLA embeddings from successful and failed rollouts, respectively. During evaluation on $\mathcal{D}_{\text{eval-seen}}$ and $\mathcal{D}_{\text{eval-unseen}}$, failure scores are

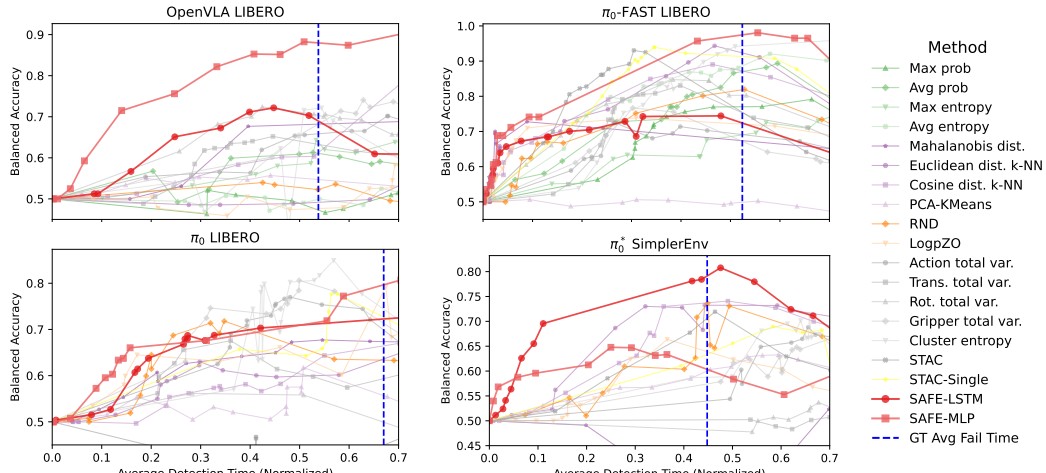

Figure 4: In all simulation experiments, the proposed *SAFE*-LSTM and *SAFE*-MLP perform better than or on par with the best baselines. The plots show the variation of balanced accuracy (bal-acc) with respect to average detection time (T-det) on $\mathcal{D}_{\text{eval-unseen}}$, under different significance levels $\alpha$ used for functional CP. Good failure detection methods should detect policy failures both accurately (high bal-acc) and proactively (lower T-det), and thus place curves towards the top left in each plot. Note that baselines in gray require multiple action samples.

computed as $s_t = d(\mathbf{e}_t, E_{\text{succ}}) - d(\mathbf{e}_t, E_{\text{fail}})$, where $d(\cdot, \cdot)$ measures the distance between a single vector and a set of vectors. Intuitively, if $\mathbf{e}_t$ is far from $E_{\text{succ}}$ and close to $E_{\text{fail}}$, it's more likely to fail. Following recent works [10, 18, 66], we ablate different types of distance, including Mahalanobis distance, and Euclidean and Cosine distance averaged over $k$-Nearest Neighbors of $\mathbf{e}_t$. We also compare to the PCA-KMeans distance measure from Liu et al. [9].

**Learned OOD Detector**: Following Xu et al. [8], we adopt LogpZO, the best-performing method, and RND [39], a strong baseline, for OOD detection–based failure detectors. Both methods use a neural network $f^{\text{OOD}}(\cdot)$ to model the embedding distribution from successful rollouts and return an OOD score for a new embedding. We adapt them to learn from both successful and failed rollouts by training two models, $f^{\text{OOD}}_{\text{succ}}(\cdot)$ and $f^{\text{OOD}}_{\text{fail}}(\cdot)$, on $E_{\text{succ}}$ and $E_{\text{fail}}$ respectively. Similar to embedding distance baselines, the failure score is computed as $s_t = f^{\text{OOD}}_{\text{succ}}(\mathbf{e}_t) - f^{\text{OOD}}_{\text{fail}}(\mathbf{e}_t)$.

**Action Consistency**: STAC [18] detects policy failures by measuring the statistical distance on the overlapping segment of two consecutive predicted action chunks. As it requires sampling multiple actions from the policy ([18] uses 256 actions), it compromises real-time operation for real robots, because unlike relatively small diffusion policy networks, large VLAs are not optimized for parallel inference[2]. Therefore, we only test STAC in the simulation experiments with 10 sampled actions. We also adopt STAC-Single, a real-time version of STAC, which computes action inconsistency using only one sample from each inference timestep. Since OpenVLA only outputs one-step immediate action ($H = 1$), STAC and STAC-single do not apply to it.

### 5.4 Evaluation Protocol

We consider two types of evaluation. The first type evaluates how well $s_t$ separates the successful and failed rollouts across all possible selections of $\delta_t$. Following the evaluation protocol widely adopted in the LLM UQ literature [22, 49, 53, 68], we use the area under the ROC curve (ROC-AUC) metric. Furthermore, because a failure flag is raised whenever $s_t$ exceeds $\delta_t$, a successful rollout (ground truth negative) becomes a false positive whenever $s_t > \delta_t$, and remains a true negative only if $s_t \leq \delta_t$ for all time. Therefore, we consider the max-so-far score $\bar{s}_t = \max_{\tau=1,\ldots,t} s_\tau$ and compute the ROC-AUC metric based on $\bar{s}_T$, the maximum failure score throughout the entire rollout.

---

[2]$\pi_0$ is 152% slower and $\pi_0$-FAST is 221% slower to generate 10 action samples compared to 1 sample, tested on a single NVIDIA RTX 3090 GPU, with vmap optimization and JiT compilation in Jax [67]. For comparison, *SAFE* methods only add negligible overhead (<1ms, or <1% of the inference time of $\pi_0$ and $\pi_0$-FAST).

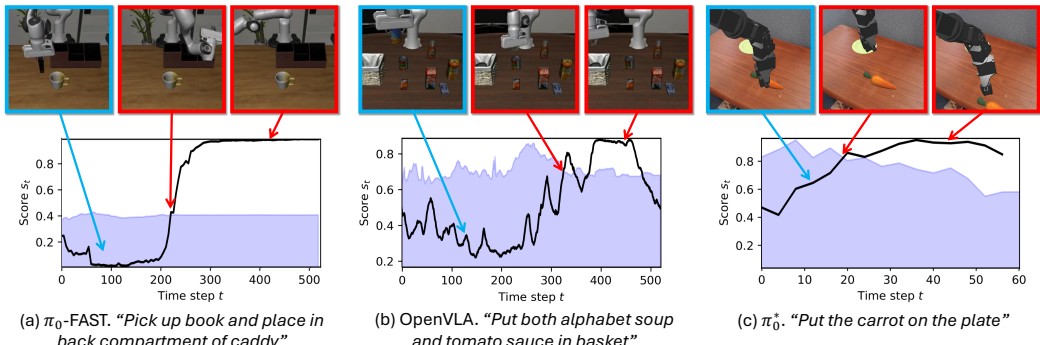

(a) $\pi_0$-FAST. *"Pick up book and place in back compartment of caddy"*

(b) OpenVLA. *"Put both alphabet soup and tomato sauce in basket"*

(c) $\pi_0^*$. *"Put the carrot on the plate"*

Figure 5: Failures detected by *SAFE*-LSTM align well with the actual robot failures, as shown in the corresponding camera observations from simulation experiments. The blue-shaded areas show the functional CP band $C_\alpha$. Once failure scores exceed $C_\alpha$, a failure flag is raised. In (a), the $\pi_0$-FAST policy misses the insertion, and its actions become unstable after that. In (b) and (c), OpenVLA and $\pi_0^*$ miss the grasp but still proceed to the placing action, causing a failure detection. Note that these tasks are not seen when training *SAFE*-LSTM.

The second type of evaluation utilizes $\delta_t = \text{upper}_t$ calibrated by functional CP in Section 4.3. By setting the significance level $\alpha$, we get a decisive positive/negative detection for each rollout. Following related works [8, 18], we consider the following metrics: true positive rate (TPR), false positive rate (FPR), balanced accuracy (bal-acc), and averaged detection time (T-det), where Bal-Acc $= \frac{\text{TPR+TNR}}{2}$. T-det is the relative timestep where $s_t > \delta_t$ for the first time (if $s_t$ never exceeds $\delta_t$, T-det becomes 1), averaged over all ground truth failed rollouts.

## 6 Results

### 6.1 How well do failure detectors distinguish failures from successes?

In Table 1 and Fig. 6 (a), we report the ROC-AUC metric based on $\bar{s}_T$, in simulation and real-world experiments, respectively. With a higher ROC-AUC metric, a failure detector achieves higher accuracy averaged over all possible thresholds. The tables show that **Token Unc.** methods have poor performance, which is aligned with findings in the LLM literature [22, 23]. On the other hand, the **Sample Consist.** and **STAC** [18] methods, which require multiple action samples, perform better and even achieve the best performance on unseen tasks in $\pi_0$-FAST LIBERO (**STAC**) and $\pi_0$ LIBERO (**Gripper total var.**). However, as these methods require multiple action samples, they cause significant overhead for VLA models and thus are not currently practical for real robots. **Embed. Distr.** methods perform well, achieving the best performance in two simulation benchmarks ($\pi_0$ and $\pi_0$-FAST) and are the second best in the real world. This demonstrates that a VLA's internal features are informative about task execution success/failure. The proposed *SAFE* methods perform better or on par with the best baselines, consistently in all settings. Averaged across simulation benchmarks, *SAFE*-**MLP** and *SAFE*-**LSTM** have similar performance, both outperforming the best baseline by 4-5% on unseen tasks, while still achieving the best performance on seen tasks. For the real-robot experiments, on both $\pi_0$-FAST+Franka and OpenVLA+WidowX rollouts, *SAFE*-**MLP** achieves the best performance and *SAFE*-**LSTM** performs closely with the best baseline (Mahala. dist. and Euclid. $k$-NN). Comparing *SAFE* with **Embed. Distr.** methods, we attribute the success of *SAFE* to its stronger ability to extract high-level abstract information from raw feature vectors through learned neural networks.

### 6.2 What is the trade-off between detection accuracy and time using functional CP?

In Fig. 4, we use $\mathcal{D}_{\text{eval-seen}}$ to calibrate the functional CP band $C_\alpha$ and evaluate on $\mathcal{D}_{\text{eval-unseen}}$. By varying the user-specified $\alpha$, we can adjust the conservativeness of the failure detectors and obtain a trade-off between accuracy (bal-acc) and detection time (T-det). A good failure detector should detect failures both accurately (higher bal-acc) and promptly (lower T-det), and thus have the curve rise toward the top-left corner in the plots of Fig. 4. As we can see from Fig. 4, the proposed *SAFE*-MLP and *SAFE*-LSTM perform the best on OpenVLA+LIBERO and $\pi_0$+SimplerEnv benchmarks, and are

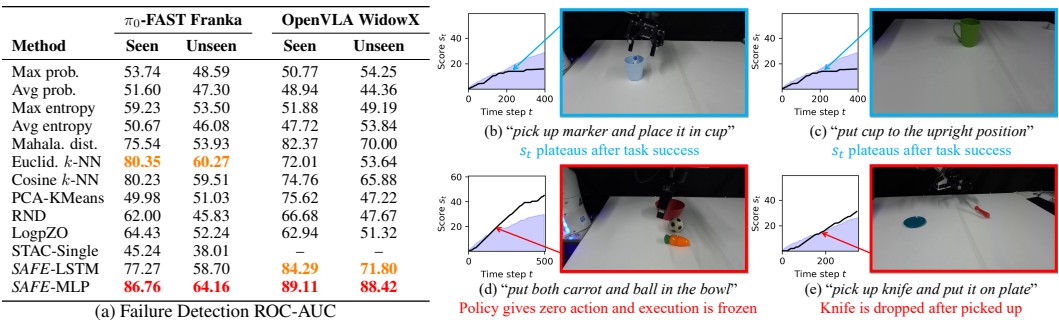

| Method | $\pi_0$-FAST Franka | | OpenVLA WidowX | |
|---|---|---|---|---|
| | **Seen** | **Unseen** | **Seen** | **Unseen** |
| Max prob. | 53.74 | 48.59 | 50.77 | 54.25 |
| Avg prob. | 51.60 | 47.30 | 48.94 | 44.36 |
| Max entropy | 59.23 | 53.50 | 51.88 | 49.19 |
| Avg entropy | 50.67 | 46.08 | 47.72 | 53.84 |
| Mahala. dist. | 75.54 | 53.93 | 82.37 | 70.00 |
| Euclid. $k$-NN | **80.35** | **60.27** | 72.01 | 53.64 |
| Cosine $k$-NN | 80.23 | 59.51 | 74.76 | 65.88 |
| PCA-KMeans | 49.98 | 51.03 | 75.62 | 47.22 |
| RND | 62.00 | 45.83 | 66.68 | 47.67 |
| LogpZO | 64.43 | 52.24 | 62.94 | 51.32 |
| STAC-Single | 45.24 | 38.01 | – | – |
| *SAFE*-LSTM | 77.27 | 58.70 | 84.29 | 71.80 |
| *SAFE*-MLP | **86.76** | **64.16** | **89.11** | **88.42** |

(a) Failure Detection ROC-AUC

(b) "*pick up marker and place it in cup*"
$s_t$ plateaus after task success

(c) "*put cup to the upright position*"
$s_t$ plateaus after task success

(d) "*put both carrot and ball in the bowl*"
Policy gives zero action and execution is frozen

(e) "*pick up knife and put it on plate*"
Knife is dropped after picked up

Figure 6: *SAFE*-MLP achieves the best failure detection performance in real-world experiments with both $\pi_0$-FAST Franka and OpenVLA WidowX. Plot (a) presents quantitative results, while (b–e) show qualitative examples from *SAFE*-MLP on the real robot. ROC-AUC values are averaged over five random seeds with different task splits.

on par with the best baseline on the other two benchmarks. We also manually annotate the ground truth (GT) failure timesteps (when a human thinks that failure happens or intervention is needed) for failed rollouts, and plot them as blue vertical lines in Fig. 4. Comparing *SAFE*'s performance with the GT fail time, we can see that *SAFE* can detect failures with high accuracy in the early stages of rollouts and potentially before the failure happens. This early detection allows early intervention for policy failures before they get stuck in execution or cause harm to the real-world environment.

### 6.3 What failure modes are detected, and do they align with human judgment?

In Fig. 5 and Fig. 6(b-d), we visualize rollouts with the failure scores detected by *SAFE*. Fig. 5 demonstrates common failure modes in simulation, including imprecise insertion, oscillatory motions, and missed grasps. Two successful rollouts on the real robot are shown in Fig. 6(b-c), where failure scores stop increasing after task completion. For the failed rollouts, the failure flag is raised after the policy is frozen (Fig. 6d) or the object slips out of the gripper (Fig. 6e). This aligns well with human intuition. Please refer to Appendix for video illustrations.

### 6.4 How efficient and practical it is to deploy *SAFE*?

*SAFE* uses a 1-2 layer MLP or LSTM and poses a minimal (less than 1%) computational overhead at runtime. For example, *SAFE*-LSTM contains 2.3 million parameters and introduces an additional 0.73 ms of inference time. This is negligible compared to large VLA models. For instance, pi0 has 3.3 billion parameters and an inference time of 149 ms. *SAFE* only requires access to the latent features of VLA models and is compatible with any white-box robot policies based on neural networks. However, SAFE does require deploying the policy and collecting successful and failed rollouts to train the failure detector before it can detect failures.

## 7 Conclusion

In this paper, we introduce the multitask failure detection problem for generalist VLA policies, where failure detectors are trained only on seen tasks and evaluated on unseen tasks. We analyze VLA's internal feature space and find that the internal features are separated for successful and failed rollouts. Based on this observation, we propose *SAFE*, a simple and efficient failure detection method by operating on the VLA's internal features. *SAFE* is evaluated on multiple VLAs in both simulation and the real world, and compared with diverse baselines. Experiments show that *SAFE* achieves SOTA results in failure detection, and aligns with human intuition.

**Limitations:** Most recent VLAs have shown capabilities in handling diverse modalities, controlling diverse embodiments, and learning latent actions from non-robotic action-less video data [69, 70]. This paper only considers multitask failure detection for manipulation tasks, and it is not clear how well the failure detectors generalize across embodiments, sim2real or to action-less videos. Besides, *SAFE* only uses features from the last layer, and how to effectively aggregate information across multiple layers of a VLA remains an open question for future work.

## Acknowledgments and Disclosure of Funding

The authors were partially supported by the Toyota Research Institute (TRI) and NSERC Discovery Grant. The authors thank Blerim Abdullai, Sebastian Aegidius, Jasper Gerigk, Ruthrash Hari, and Wei-Cheng Tseng for helpful discussions and feedback. The authors also thank the anonymous reviewers and area chairs for their constructive feedback.

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

# Appendix

## A  Potential Societal Impact

This work advances the safety and reliability of VLAs through multitask failure detection, which can reduce unintended behaviors during robot deployment. However, the proposed framework could be misused in surveillance or fully autonomous systems with limited human oversight. Additionally, biases or privacy issues may arise from training data collected during robot interactions. These risks can be mitigated through responsible data handling, transparency in model release, and maintaining human oversight in downstream applications.

## B  Experiment Details

### B.1  Vision-Language-Action Models

We conduct experiments on 3 state-of-the-art large VLA models: OpenVLA [2], $\pi_0$ [4] and $\pi_0$-FAST [5]. Given the internal feature vectors $E \in \mathbb{R}^{n \times d'}$ produced by a VLA model, where dimension $n$ corresponds to different token positions, diffusion steps, etc. and $d$ is the feature dimension, we aggregate $E$ into a single fixed-dimensional feature vector $\mathbf{e} \in \mathbb{R}^d$ before inputting to the proposed *SAFE* models. In this paper, we consider and ablate the following ways of feature aggregation:

- First: take the first vector along the dimension $n$, $\mathbf{e} = E_1$;
- Last: take the last vector along the dimension $n$, $\mathbf{e} = E_n$;
- Mean: take average over the dimension $n$, $\mathbf{e} = \frac{1}{n} \sum_{i=1}^n E_i$;
- First&Last: concatenate the first and the last vector, $\mathbf{e} = \text{concat}(E_0, E_n) \in \mathbb{R}^{2d'}$.

Both OpenVLA and $\pi_0$-FAST first predict a sequence of discrete tokens and then convert them into continuous actions. We take feature vectors before being decoded into the output tokens from the last transformer block as $E \in \mathbb{R}^{n \times d'}$, and therefore $n$ corresponds to the number of generated tokens. We ablate the aggregation method along this dimension and denote the aggregation method as $agg_{\text{token}}$. For $\pi_0$-FAST, we additionally ablate using the feature vectors before ("encoded") and after ("pre-logits") the final RMS normalization layer as $E$.

Differently, $\pi_0$ (and $\pi_0^*$) outputs action vectors by flow matching [71], and we take the feature vectors before being projected into the velocity field. Suppose $\pi_0$ predict an action chunk of horizon $H$ and performs $k$ flow matching steps, the internal features become $E \in \mathbb{R}^{H \times k \times d}$ and we perform the aggregation process along the $H$ dimension and the $k$ dimension separately to get the final embedding vector $\mathbf{e} \in \mathbb{R}^{d'}$. The aggregation methods are denoted as $agg_{\text{hori}}$ and $agg_{\text{diff}}$ along these two dimensions, respectively.

For all VLA models, we ablate different methods of aggregating the hidden features $E$ into a single feature vector $\mathbf{e}$ and select the best method according to $\mathcal{D}_{\text{eval-seen}}$ performance. The detailed ablation results are shown in Appendix B.8.

OpenVLA and $\pi_0^*$ use MIT License; $\pi_0$ and $\pi_0$-FAST use Apache-2.0 license.

### B.2  *SAFE* Failure Detector

*SAFE*-LSTM uses an LSTM model with 1 layer and a hidden dimension of 256, and an additional linear layer is used to project the hidden states of LSTM into a single scalar $s_t$. *SAFE*-MLP uses a multi-layer perceptron with 2 layers and a hidden dimension of 256. Since successes and failures from the generated rollouts are imbalanced, the losses on positive (failed) and negative (successful) rollouts are weighted by their inverse class frequency. We also apply an L2 regularization loss on the model weights to reduce overfitting, and this loss is weighted by $\lambda_{\text{reg}}$ and optimized together with the failure score learning loss $L_{\text{LSTM}}$ or $L_{\text{MLP}}$. $\lambda_{\text{reg}}$ are determined by grid search.

### B.3 Failure Detection Baselines

For the cluster entropy baseline, we use agglomerative clustering with the ward linkage criterion [72]. The distance threshold is denoted as $\delta$ and decided by grid search.

For the STAC baseline [18], we use the Maximum Mean Discrepancy (MMD) distance measure [73] with radial basis function kernels, which was reported to have the best performance by [18]. The bandwidth of the RBF kernel is 1.

For all baselines except for RND [39] and LogpZO [8], we ablate one version that only considers the failure score computed from the current timestep ("cumsum=False") and another that uses the cumulative sum (cumsum) of the failure scores over time ("cumsum=True").

For RND and LogpZO, we use the original implementation provided by the authors [3] and do not accumulate scores. In [8], RND and LogpZO are trained to model the distribution of (encoded) observations $\mathbf{o}_t$ and predicted actions $\mathbf{A}_t$. In this work, we adapt them to model the distribution of VLA's internal embeddings $\mathbf{e}_t$.

Note that as $\pi_0$ (and $\pi_0^*$) does not output discrete tokens, token uncertainty-based baselines do not apply. And for OpenVLA, $H = H' = 1$ and thus the STAC [18] and STAC-Single do not apply.

### B.4 Conformal Prediction

We follow [8, 21] for CP band construction. Please refer to Section. B in the Appendix of [8] for a detailed formulation. Specifically, in our experiments, we use the adaptive modulation function (Equation 2 in the Appendix of [8]), which models the non-extreme behaviors of the functional data.

### B.5 Benchmark Details

**LIBERO** [56]: We adopt the LIBERO-10 task suite, which contains the most diverse set of objects, environments, and instructions among the 4 LIBERO task suites, and therefore LIBERO-10 is considered the most challenging task suite. LIBERO-10 contains 10 tasks with 50 rollouts in each task. We use the initial conditions for all rollouts as specified and provided by the author[4]. To test VLA models on LIBERO, we adopt the trained model weights provided by the respective authors and do not further finetune them. On LIBERO-10, OpenVLA achieves a success rate of 53.7%, $\pi_0$-FAST achieves 60.2%, and $\pi_0$ achieves 85.2%. For evaluation, 3 out of 10 tasks are unseen, and within seen tasks, 60% of rollouts are used for $\mathcal{D}_{\text{train}}$ and the remaining 40% for $\mathcal{D}_{\text{eval-seen}}$.

Note that the LIBERO simulator stops the rollout execution when the robot finishes the task (considered a success) or a maximum rollout length is reached (considered a failure). Therefore, in the generated rollouts, failed ones always have the maximum length, but successful ones are shorter. This could result in an unfair advantage for some of the compared failure detectors (if a failure detector simply learns to count the time elapsed, i.e., $s_t = t$, it will achieve perfect failure detection since failed rollouts have a fixed and longer duration). To ensure a fair comparison, for evaluation in Table 1, we compute the minimum rollout length for each task and use that as $T$ for that task. The failure detection performance (in ROC-AUC) is then determined based on $s_T$, where $T$ is the same for all successful and failed rollouts within each task.

LIBERO benchmark uses the MIT license.

**SimplerEnv** [63]: SimplerEnv carefully identifies and reduces the domain gap between the simulation and the real-world demonstration data, and provides simulated environments that highly resemble the demonstration data from RT-series [1, 7, 31] (with the Google Robot embodiment) and BridgeData V2 [33] (with the WidowX embodiment). They show that models pretrained on real-world datasets can also accomplish similar tasks in SimplerEnv without finetuning, and their performance in simulation matches that in the real world.

On this benchmark, we adopt the pretrained model checkpoints of $\pi_0^*$ [64]. Note that $\pi_0^*$ model checkpoints are trained separately on the Google Robot embodiment and the WidowX embodiment, which results in two model checkpoints that have different internal feature spaces. Therefore, all

---

[3]https://github.com/CXU-TRI/FAIL-Detect
[4]https://github.com/Lifelong-Robot-Learning/LIBERO/tree/master/libero/libero/init_files

Table 2: List of tasks used in $\pi_0^*$ + SimplerEnv benchmark.

| Embodiment | Task ID | Environment Name | $\pi_0^*$ Success Rate (%) |
|---|---|---|---|
| Google Robot | 1 | `google_robot_move_near_v0` | 77 |
| Google Robot | 2 | `google_robot_open_drawer` | 50 |
| Google Robot | 3 | `google_robot_close_drawer` | 80 |
| Google Robot | 4 | `google_robot_place_apple_in_closed_top_drawer` | 40 |
| WidowX | 1 | `widowx_carrot_on_plate` | 44 |
| WidowX | 2 | `widowx_put_eggplant_in_basket` | 88 |
| WidowX | 3 | `widowx_spoon_on_towel` | 79 |
| WidowX | 4 | `widowx_stack_cube` | 43 |

Table 3: List of tasks used in the real-world Franka experiments.

| Task | Instruction | Rollout Length $T$ |
|---|---|---|
| 1 | close the door | 300 |
| 2 | close the drawer | 200 |
| 3 | pick up the ball and place it in the bowl | 400 |
| 4 | pick up the knife and put it on the plate | 350 |
| 5 | pick up the lid and place it on the pot | 400 |
| 6 | pick up the lid from the pot and place it on the table | 400 |
| 7 | pick up the marker and place it in the cup | 400 |
| 8 | place the green block on the yellow block | 350 |
| 9 | place the pink cup to the right of the blue cup | 300 |
| 10 | press the button | 200 |
| 11 | put both the carrot and the ball in the bowl | 500 |
| 12 | put the cup to the upright position | 500 |
| 13 | unfold the cloth | 500 |

failure detectors are trained and evaluated on each embodiment separately as well. All reported evaluation metrics are computed separately for each embodiment and then averaged. In Table 2, we list the tasks used for failure detection on SimplerEnv. We generate 100 rollouts for each task with random initial configurations, and the success rates of $\pi_0^*$ on each task are also listed in Table 2. A rollout stops after the maximum number of allowed timesteps have passed, regardless of task success or failure. Within each embodiment, 1 out of 4 tasks is unseen, and within the seen tasks, 66% of the rollouts are in $\mathcal{D}_{\text{train}}$ and the remaining 33% in $\mathcal{D}_{\text{eval-seen}}$.

SimplerEnv benchmark uses the MIT license.

**Real-world experiments with Franka robot**: In Table 3, we list the tasks used in the real-world experiments. For each task, we set a number of timesteps $T$ allowed for one rollout, and all rollouts of the same task are terminated after the same $T$ timesteps regardless of task success or failure. In Fig. 7, we further visualize some example successful and failed rollouts from the real-world experiments.

**Real-world experiments with WidowX robot**: We also tested the OpenVLA model pretrained on the "Open-X Magic Soup++" dataset on a real WidowX robot arm in our lab. In this experiment, we collected a total of 532 rollouts on the following 8 tasks (listed in Table 4), with 244 successes and 288 failures. Each task has roughly the same number of rollouts.

## B.6 Benchmark Statistics

In the Table 5, we summarize the statistics on the number of tasks and rollouts collected for each benchmark and how they are split into training and evaluation subsets. We note that as SAFE is designed for multitask failure detection, it is trained on only a limited set of training tasks and rollouts and can generalize to new tasks without further collecting rollouts. While SAFE does require hundreds of rollouts from multiple tasks during training, when handling new tasks, SAFE becomes more efficient than existing task-specific failure detectors (like [8, 18]) that require collecting rollouts for training and calibration for every new task encountered.

Table 4: List of tasks used in the real-world experiments.

| Task | Instruction |
|------|-------------|
| 1 | Lift AAA Battery |
| 2 | Lift Eggplant |
| 3 | Lift Red Bottle |
| 4 | Lift Blue Cup |
| 5 | Put Blue Cup on Plate |
| 6 | Put the Red Bottle into Pot |
| 7 | Put the Carrot on Plate |
| 8 | Put the Red Block into the Pot |

| Benchmark | Number of Tasks | | | Number of rollouts | | | |
|-----------|------|--------|-------|-------|-----------|-------------|-------|
| | Seen | Unseen | Total | Train | Eval Seen | Eval Unseen | Total |
| LIBERO | 7 | 3 | 10 | 210 | 140 | 150 | 500 |
| $\pi_0^*$ SimplerEnv, Google Robot | 2 | 2 | 4 | 198 | 102 | 100 | 400 |
| $\pi_0^*$ SimplerEnv, WidowX | 2 | 2 | 4 | 198 | 102 | 100 | 400 |
| Octo SimplerEnv | 9 | 3 | 12 | 594 | 306 | 300 | 1200 |
| Real Franka | 10 | 3 | 13 | 450 | 150 | 180 | 780 |
| Real WidowX | 6 | 2 | 8 | 250 | 133 | 149 | 532 |

Table 5: Benchmark statistics for how tasks and rollouts are split into different subsets. Note that as we preformed multiple runs with different random seeds for all experiments, each run will use different set of tasks for seen and unseen subsets.

### B.7  Training Details

We use Adam optimizer [74] with $\beta_1 = 0.9$, $\beta_1 = 0.999$, $\epsilon = 10^{-8}$, and a learning rate (lr) determined by grid search. The *SAFE* models are trained for 1000 epochs with batch size 512. Note that each rollout is considered as one data point and thus batch size of 512 translates to training on (at most) 512 rollouts in each iteration. All training and evaluation are done on a single NVIDIA A100 40GB GPU. Since *SAFE* uses small networks (MLP or LSTM with 1 or 2 layers), the typical training time for one model is less than one minute.

### B.8  Hyperparameter Tuning

To determine the hyperparameters for the proposed *SAFE* and baselines, we perform a grid search over them and select the ones with the highest failure detection performance (ROC-AUC) on the $\mathcal{D}_{\text{eval-seen}}$ split. In Table 9, Table 10, and Table 11, we report the hyperparameters we have searched over and the values with the best performance. Note that for the real-world experiments, we fix the $\mathbf{e}_t$ to be the "pre-logits" with "Mean" aggregation.

## C  Additional Results

### C.1  Feature Visualization and Analysis

We perform the feature analysis similar to Section 4.1 and Fig. 1 on other benchmarks and show the plots in Fig. 8, Note that in this feature analysis process, the t-SNE algorithm was performed on the VLA's embeddings without any learning. Therefore, the feature dimension reduction process is unsupervised and does not know about task successes or failures.

Comparing the plots in Fig. 1 and Fig. 8, we can see that the embedding spaces from VLAs are different from each other, which corresponds to the different failure patterns presented by the VLAs. For $\pi_0$-**FAST** (Fig. 1) and $\pi_0$ **on LIBERO** (Fig. 8a and b), when task execution fails, the embeddings fall into the same region ("failure zone"). This corresponds to the major failure mode of $\pi_0$-FAST trained on the LIBERO dataset, where the predicted actions $\mathbf{A}_t$ become unstable and the robot arms move to weird configurations and out of the observation frame. For **OpenVLA on LIBERO** (Fig. 8c and d). we observe that for most failed rollouts, the robot freezes at or shakes around certain

configurations during the middle of task execution. Such failed rollouts result in features very close to each other, which corresponds to small blobs of red dots in Fig. 8c.

Despite the different appearances of the embedding spaces from the above benchmarks, their successful and failed rollouts are separable in the feature space. This is aligned with the high performance of the proposed *SAFE* and the embedding-based baseline methods. Moreover, although the embeddings of the failed rollouts from OpenVLA are spread over the space and do not form a unified "failure zone", *SAFE* is still able to learn to separate task failures from successes (possibly by extracting the correlations that are not visualized by t-SNE) and generalize well to unseen tasks, as reported in Table 1.

However, the visualized embeddings of $\pi_0$-**FAST on the real Franka robot** (Fig. 8e and f) are different, where embeddings from successful and failed rollouts are not easily separable through the t-SNE visualization. We hypothesize that because the tasks we used for real-world experiments are more diverse, their failures do not have a unified semantic meaning, and thus the embeddings are not clearly separated in the visualization. This explains the limited performance of all failure detection methods as reported by Fig. 6, where ROC-AUC is at most 64 on $\mathcal{D}_{\text{eval-unseen}}$. Nevertheless, *SAFE*-MLP still outperforms all baselines on both seen and unseen splits in this evaluation.

## C.2 Conformal Prediction Results

We use functional CP [8, 21] to determine the time-varying thresholds $\delta_t$ for failure detection. By varying the significance level $\alpha$ used in functional CP, we can adjust the conservativeness of failure detection and get different performance. In Fig. 9, we plot the change of TNR (True Negative Rate), TPR (True Positive Rate) and Bal-acc (Balanced Accuracy, $\frac{\text{TNR}+\text{TPR}}{2}$) w.r.t. $\alpha$. From Fig. 9, we can observe that while the metrics do vary with the $\alpha$, choosing $\alpha = 0.15$ (or in general, between 0.05 and 0.2) performs well across the board. We have also chosen $\alpha$ to be 0.15 for most qualitative results and analyses reported in the paper.

Note that we calibrate the CP bands on successful rollouts (negative data points), and thus if the assumptions used in CP ($s_t$ are sampled i.i.d.) hold, the TNR rate is lower bounded by and close to $1 - \alpha$ (the gray dashed line in the TNR plots in Fig. 9) [21]. However, as the multitask failure detection problem requires detecting failures on tasks that are not in the training or the calibration sets, we need to calibrate CP bands on $\mathcal{D}_{\text{eval-seen}}$ and then evaluate them on $\mathcal{D}_{\text{eval-unseen}}$. Therefore, the i.i.d assumption may not hold, and TNR may deviate from the gray dashed line.

From Fig. 9, we can see that on OpenVLA+LIBERO and $\pi_0^*$+SimplerEnv benchmarks, the TNR curves obtained by *SAFE* are close to the gray dashed line $1 - \alpha$, while those on the other 3 benchmarks are lower than $1 - \alpha$. A similar phenomenon is also observed for the baseline methods: none of the TNR curves obtained from the baselines consistently conform to the $1 - \alpha$ curve across all benchmarks. We attribute this to the challenging nature of the multitask failure detection problem, where the failure scores for calibration and evaluation may not come from the same distribution. Nevertheless, we still adopt the functional CP as a principled method to determine the time-varying failure detection threshold $\delta_t$. Moreover, from Fig. 9, we can see that *SAFE* can achieve higher TPR and result in fewer false negatives compared to the baselines. This is crucial for safety-critical environments, where a missing failure (false negative) can be much more catastrophic than a false alarm (false positive).

## C.3 Failure Detection Time

As mentioned in Section 6.2, we manually label when the failure happens for the failed rollouts. The labeling process is based on video recordings after all rollouts are collected and no interventions were done during the task execution. While the exact times of failure are clear for some failure modes (e.g. dangerous actions, breaking objects), they can be ambiguous and hard to annotate for other failure modes. For example, a policy may freeze in the middle of task execution, and after that either recovering from it or getting stuck indefinitely can be possible. In another case, a policy may repeatedly try grasping the object but keep missing the grasp until timeout, and it's hard to determine a single point of failure. To handle such cases, we instruct the human annotators to pick the time where they think intervention is needed and they should take over control to prevent an execution failure. In practice, for the above ambiguous failure modes, we annotate the failures after the policy gets stuck by a few seconds or re-tries the grasping action a few times. For some rollouts that look

Table 6: Performance on the OpenVLA+LIBERO benchmark using different numbers of training tasks.

| # Training Tasks | 1 | | 3 | | 5 | | 7 | |
|---|---|---|---|---|---|---|---|---|
| **Eval Task Split** | Seen | Unseen | Seen | Unseen | Seen | Unseen | Seen | Unseen |
| Mahalanobis | 40.21 | 52.75 | 58.00 | 52.31 | 57.68 | 50.78 | 62.03 | 58.85 |
| Euclid. $k$-NN | 49.74 | 63.76 | 61.66 | 67.02 | 59.14 | 67.11 | 66.00 | 55.23 |
| Cosine. $k$-NN | 53.27 | 60.76 | 65.39 | 65.64 | 67.46 | 70.57 | 67.09 | 69.45 |
| PCA-KMeans | 60.39 | 40.58 | 61.18 | 52.87 | 61.50 | 53.06 | 57.18 | 55.10 |
| RND | 29.29 | 50.32 | 54.46 | 47.39 | 56.71 | 49.15 | 52.57 | 46.88 |
| LogpZO | 61.75 | 56.17 | 52.89 | 50.49 | 65.99 | 56.60 | 61.57 | 52.91 |
| SAFE-LSTM | 50.88 | 52.25 | 68.85 | 63.31 | 70.70 | 66.31 | 70.24 | 72.47 |
| SAFE-MLP | 54.34 | 63.76 | 67.86 | 67.03 | 69.32 | 68.17 | 72.68 | 73.47 |

very plausible but do not succeed due to the time limit, the failure time is annotated as the end of the rollout. Note that we annotate only the failed rollouts and not the successful ones, even though they may also show subtle signs of failure in the middle.

In Fig. 10, we compare the times of failure detected by the proposed *SAFE*-MLP model and a human annotator. From Fig. 10, we can see that for both $\pi_0$ and $\pi_0$-FAST models, *SAFE*-MLP can detect failures before they happen (as identified by a human). When used for $\pi_0$-FAST deployed on LIBERO, *SAFE*-MLP can *forecast* failures well in advance and even predict 40% of the failures after the first timestep.

Furthermore, from Fig. 10a and Fig. 10c, we can see that the blue curves jump up on the right edge of the plots. This means that the human annotator does not think these rollouts are failures until the very last moment, where the VLA model is probably on the right track and fails only due to timeout. We think such failures are also hard for failure detectors to detect, and it explains the low performance of all failure detectors on these benchmarks.

### C.4 Result Variance

In Table 8, we report the standard deviation for all results in Table 1 and Fig. 6 left. Note that for the repeated runs, not only are they using different random seeds, also the tasks are split differently into the seen and the unseen subsets. Since different tasks have different difficulties for failure detection, it is normal to see large standard deviations in Table 8. From Table 8, we can see that the proposed *SAFE* methods achieve high averaged performance with relatively low standard deviations compared to the baselines, across all evaluation benchmarks. This signifies the strong and also stable performance of *SAFE*.

## D  Additional Ablation Studies

### D.1  Number of Training Tasks

As SAFE learns to distinguish failures from successes from training rollouts, the diversity of failure modes and the number of tasks in the training data have an effect on the failure detection performance. To quantify this effect, we conduct an experiment varying the number of seen tasks that are used in training. Note that different tasks typically also have different failure modes, and in this way, we are also ablating the diversity of failure modes.

In Table 6, we report the failure detection ROC-AUC on the OpenVLA+LIBERO benchmark, trained on different numbers of tasks. While the number of seen tasks is ablated, all experiments use the same set of unseen tasks for evaluation, and performance on unseen tasks is comparable. All numbers are averaged over 3 random seeds. Experiments with 7 tasks for training match the setting reported in the paper. Training-free methods do not depend on training tasks and are not shown.

Table 6 shows that for most methods, with more tasks used for training, the performance on unseen tasks gets better. The proposed SAFE-MLP performs well in all settings and can also achieve good performance when fewer (3 or 5) tasks are used for training.

Table 7: Comparison of model performance across different visual encoders and architectures.

| Method | LSTM | | MLP | |
|---|---|---|---|---|
| Eval Task Split | Seen | Unseen | Seen | Unseen |
| DINOv2 | 76.93 | 56.96 | 76.20 | 59.46 |
| CLIP | 76.77 | 52.71 | 77.88 | 59.77 |
| DINOv2+CLIP | 77.09 | 59.65 | 76.36 | 58.43 |
| VLA (Ours) | 77.27 | 58.70 | 86.76 | 64.16 |

## D.2 Features from Foundation Models

In Table 7, we ablate SAFE-MLP and SAFE-LSTM using DINOv2 features, CLIP features, DINOv2 and CLIP concatenated (DINOv2+CLIP), and the VLA last-layer features (VLA; our main method). DINOv2 and CLIP features are extracted from the observation images, and this experiment is conducted over the real-world Franka rollouts, following the same setting as reported in the paper. Numbers are averaged ROC-AUC on the Seen and Unseen tasks.

The best performing method in the above table is the SAFE-MLP method based on VLA last-layer features, where VLA features outperform other feature types by a large margin. We think that this is because VLA feature space learns high-level information about the tasks, and thus can more easily distinguish failures from successes than general pretrained models. Similar findings were also reported in related works like [18].

# E   Additional Discussions

## E.1   Comparing Failure Detection, Uncertainty Quantification and OOD Detection

Failure detection, uncertainty quantification and OOD detection are three closely connected concepts with subtle differences. *SAFE* learns to model the probability of failures and detect failures of VLA policies, but it achieves this not through uncertainty quantification (UQ) or OOD detection. Here, we provide a detailed discussion comparing these three concepts.

**Failure detection** is the task of detecting failures when a robot is performing certain tasks. *SAFE* learns the likelihood of failure through training on a set of successful and failed rollouts. *SAFE*-LSTM is trained by BCE loss and outputs a normalized score indicating the probability of failure of VLA. The output of *SAFE*-MLP is not normalized and thus not a probability. However, output scores from both *SAFE*-LSTM and *SAFE*-MLP are calibrated through functional Conformal Prediction CP and can be used for failure detection with theoretical guarantees.

**Uncertainty quantification** (UQ) measures a VLA's uncertainty in its outputs and can be used as a proxy for failure detection. In our experiments, the token uncertainty baselines and sample consistency baselines are inspired by LLM/VLM literature and designed based on UQ. Methods in this category are typically training-free, but they only show limited success according to our experiments.

**OOD detection**-based failure detection methods treat successful rollouts as normal execution conditions and assume that deviations from this norm lead to a higher chance of failure. In our experiments, the embedding distribution-based baselines are designed to detect policy failure based on OOD detection. Methods in this category can work without failed rollouts. In our experiments, we adapted them to take in both successes and failures, and they showed strong performance. Please see Section 2.2 in our paper and also [8] for more comprehensive discussions on these methods.

Uncertainty quantification methods have been widely used for LLM/VLM hallucination detection (see Section 2.3), and OOD detection-based methods have been shown to be effective for failure detection in robotics policies (see Section 2.2). Therefore, we think it's appropriate to use them as baselines.

Different from existing works based on UQ or OOD detection, *SAFE* directly learns to detect failures from a history of observations and the language instruction specifying the desired task without using

uncertainty or OOD detection as the proxy measurement. Experiments show that this direct learning regime used by *SAFE* is more effective and achieves better performance than other methods.

# F  Potential Future Works

## F.1  Extending Beyond the Last-layer features

In this paper, we maximize the simplicity and transferability when designing *SAFE*. By only taking feature vectors at the last layer, the proposed method can be easily integrated into any VLA models with minimal implementation changes and no finetuning on the VLAs themselves.

Further fusing or aggregating deep features from multiple layers can also benefit failure detection and is a promising future direction. Related works have shown potential in this direction. For example, [55] proposed Truthfulness Separator Vector (TSV), which is injected in the LLM latent features in the middle of the transformer and is optimized to better separate the hallucinated and truthful responses in the final token feature space. We think a similar technique can also be developed for VLA failure detection. However, this would require a special design and implementation for each VLA model (as some VLA models output discrete tokens, and others use flow matching to output continuous actions, there may not be a single design that can be applied to all VLA architectures), reducing its transferability. We leave the development of such methods as a promising future work.

Using the latent feature from an intermediate transformer block may also be a promising future direction. As shown by [49] and [53], LLM latent features from different layers have different performance on hallucination detection, and the best one may not be the last layer. However, exactly which layer works the best may depend on the model and require extensive ablation experiments to find out. For example, as reported by [49], for the OPT-6.7B model, the 20th layer works the best, but for LLAMA2-7B, the 16th layer works the best. To locate the best layer, [49] has to perform a grid search over each LLM tested. On the contrary, in our setting, VLA users can avoid such grid search experiments and simply choose the final layer for failure detection. Therefore, we think that precisely finding the layer that works the best for VLA failure detection is outside the scope of this paper, but it would be very interesting to explore for future work.

## F.2  Adaptive Thresholding by Online Conformal Prediction

The proposed *SAFE* and baselines can be extended to online or adaptive conformal prediction (CP) [75]. In such a framework, rollout results are observed one-by-one and compared to the prediction results, and the significance level $\alpha$ is adjusted for each individual task if the prediction is wrong. However, when VLA policies are deployed, they may constantly meet novel environments and customized task instructions, and may rarely repeat the same task. In such a case, it may be less pratical to develop adaptive CP band for each specific task. Therefore, in this work, we focus on the performance of failure detectors when they are directly deployed on a novel task, and have never seen or repeated the task before. In this setting, offline CP is more appropriate. Nevertheless, we believe online CP is an interesting extension to our work, and we leave it as an important future work.

## F.3  Using Detected Failures for Behavior Improvement

*SAFE* focuses on detecting failures accurately and in a timely manner, which enables either the robot to abort potentially dangerous actions or a human monitor to step in and take over control. How to learn a recovery policy or how to improve the VLAs themselves are important areas for future work. In this work, we focused on detecting failures of multitask VLAs reliably and in real-time, which is a crucial stepping stone towards autonomous recovery (e.g., with a fallback policy) and policy improvement (e.g., through interactive imitation learning).

We think it is possible to use the findings from this paper to further develop methods for steering or improving VLA behaviors. For example, we show that the embeddings for successful and failed rollouts are separated in the latent space, so it is possible to learn a steering vector that manipulates the latent activations of a VLA and changes its output actions, as done in [50] and [76]. However, different from the stylization or hallucination reduction tasks for LLMs, robot manipulation involves multistep closed-loop interaction between the policy and the environment, which greatly complicates the relationship between VLA outputs and task execution successes. Therefore, how to improve

Table 8: Mean and standard deviation of failure detection ROC-AUC on all benchmarks. This table complements the results from Table 1 and Fig. 6 left.

| VLA Model | OpenVLA | | $\pi_0$-FAST | | $\pi_0$ | | $\pi_0^*$ | | $\pi_0$-FAST | |
| Benchmark | LIBERO | | LIBERO | | LIBERO | | SimplerEnv | | Real Franka | |
| Eval Task Split | Seen | Unseen | Seen | Unseen | Seen | Unseen | Seen | Unseen | Seen | Unseen |
|---|---|---|---|---|---|---|---|---|---|---|
| Max prob. | 50.25±2.51 | 53.83±6.32 | 61.32±9.57 | 69.44±13.61 | - | - | - | - | 53.74±3.46 | 48.59±3.00 |
| Avg prob. | 44.05±1.26 | 51.58±1.82 | 52.46±3.44 | 58.04±5.64 | | | | | 51.60±3.12 | 47.30±4.32 |
| Max entropy | 52.94±4.36 | 53.09±7.68 | 46.69±13.33 | 62.96±19.62 | | | | | 59.23±3.06 | 53.50±3.15 |
| Avg entropy | 45.27±1.78 | 50.03±3.18 | 50.93±1.22 | 58.63±3.47 | - | - | | | 50.67±3.96 | 46.08±4.79 |
| Mahalanobis dist. | 62.03±5.11 | 58.85±4.16 | 93.56±2.32 | 83.79±7.18 | 77.12±8.57 | 74.31±12.64 | 88.42±2.82 | 52.84±31.97 | 75.54±4.07 | 53.93±5.06 |
| Euclidean dist. $k$-NN | 66.00±2.33 | 55.23±10.05 | 92.04±2.39 | 84.12±6.47 | 75.64±6.20 | 70.73±16.69 | 89.73±3.08 | 68.41±9.22 | 80.35±5.36 | 60.27±4.79 |
| Cosine dist. $k$-NN | 67.09±2.74 | 69.45±6.14 | 92.09±1.70 | 84.64±4.90 | 75.76±6.16 | 70.31±16.84 | 90.19±4.05 | 71.32±12.02 | 80.23±5.12 | 59.51±5.76 |
| PCA-KMeans [9] | 57.18±2.04 | 55.10±1.16 | 68.46±4.92 | 57.12±10.44 | 64.92±8.90 | 60.35±19.93 | 66.88±5.10 | 61.19±14.76 | 51.91±4.20 | 49.86±6.19 |
| RND [39] | 52.57±4.56 | 46.88±4.92 | 88.67±3.05 | 81.57±8.67 | 71.92±7.02 | 69.44±19.39 | 85.07±4.04 | 65.89±6.52 | 62.00±5.44 | 45.83±5.10 |
| LogpZO [8] | 61.57±3.62 | 52.91±5.79 | 91.52±2.39 | 83.07±7.17 | 76.80±9.12 | 73.23±11.64 | 88.79±4.92 | 74.66±14.96 | 64.43±7.82 | 52.24±3.68 |
| Action total var. | 62.76±1.66 | 65.43±2.50 | 76.95±7.22 | 74.50±12.19 | 77.20±5.65 | 75.18±5.08 | 68.41±10.81 | 67.94±15.97 | - | - |
| Trans. total var. | 55.33±2.06 | 58.99±5.13 | 78.21±4.09 | 80.03±9.11 | 49.38±9.95 | 54.71±7.57 | 63.27±7.17 | 55.90±19.19 | | |
| Rot. total var. | 47.85±2.88 | 55.30±4.38 | 80.87±5.85 | 77.29±8.71 | 52.94±7.56 | 61.06±10.60 | 58.07±10.41 | 62.10±9.39 | | |
| Gripper total var. | 61.84±2.67 | 64.48±3.05 | 76.82±7.10 | 74.42±12.13 | 77.19±5.66 | 75.19±5.08 | 69.16±9.50 | 69.29±14.77 | | |
| Cluster entropy | 50.16±2.36 | 51.44±1.01 | 80.22±7.37 | 80.53±8.65 | 76.19±4.31 | 72.12±1.04 | 68.25±9.03 | 73.66±16.03 | - | - |
| STAC [18] | - | - | 83.07±4.61 | 85.31±6.71 | 46.55±8.90 | 47.91±20.94 | 60.74±13.89 | 62.21±16.72 | - | - |
| STAC-Single | - | - | 85.46±6.55 | 81.16±8.63 | 68.46±5.10 | 69.39±8.22 | 68.71±7.06 | 70.40±8.76 | 45.24±3.68 | 38.01±9.81 |
| *SAFE*-LSTM | 70.24±1.49 | 72.47±5.55 | 92.98±2.62 | 84.48±7.29 | 76.98±5.34 | 71.09±6.61 | 88.85±6.30 | 80.11±10.49 | 77.27±5.82 | 58.70±4.37 |
| *SAFE*-MLP | 72.68±2.38 | 73.47±5.39 | 90.06±2.82 | 80.44±5.72 | 73.50±7.43 | 73.27±11.85 | 89.50±4.49 | 84.82±8.12 | 86.76±2.64 | 64.16±5.88 |

VLAs through activation steering is a challenging and open research question beyond the scope of our paper.

Table 9: Grid-searched and best-performing hyperparameters (in bold text) for OpenVLA+LIBERO (left) and $\pi_0$-FAST+LIBERO (right).

| Method | HParams | Values |
|---|---|---|
| Max prob. | cumsum | True **False** |
| Avg prob. | cumsum | True **False** |
| Max entropy | cumsum | True **False** |
| Avg entropy | cumsum | True **False** |
| Mahalanobis dist. | $agg_{token}$ | First **Last** Mean |
| | cumsum | True **False** |
| Euclidean dist. $k$-NN | $agg_{token}$ | First Last **Mean** |
| | cumsum | **True** False |
| | $k$ | 1 5 **10** |
| Cosine dist. $k$-NN | $agg_{token}$ | First **Last** Mean |
| | cumsum | **True** False |
| | $k$ | 1 5 **10** |
| PCA-KMeans | $agg_{token}$ | First **Last** Mean |
| | cumsum | True **False** |
| | clusters | 16 32 **64** |
| | dim | 32 64 **128** |
| RND | $agg_{token}$ | **First** Last Mean |
| LogpZO | $agg_{token}$ | First Last **Mean** |
| Action total var. | cumsum | **True** False |
| Trans. total var. | cumsum | True **False** |
| Rot. total var. | cumsum | **True** False |
| Gripper total var. | cumsum | **True** False |
| Cluster entropy | cumsum | True **False** |
| | $\delta$ | **0.01** 0.05 |
| *SAFE*-LSTM | $agg_{token}$ | First **Last** Mean |
| | lr | **1e-4** 3e-4 1e-3 |
| | $\lambda_{reg}$ | 1e-3 1e-2 1e-1 **1** |
| *SAFE*-MLP | $agg_{token}$ | First **Last** Mean |
| | lr | **1e-4** 3e-4 1e-3 |
| | $\lambda_{reg}$ | 1e-3 **1e-2** 1e-1 1 |

| Method | HParams | Values |
|---|---|---|
| Max prob. | cumsum | True **False** |
| Avg prob. | cumsum | **True** False |
| Max entropy | cumsum | True **False** |
| Avg entropy | cumsum | **True** False |
| Mahalanobis dist. | $agg_{token}$ | First Last **Mean** |
| | Feat | Encoded **Pre-logits** |
| | cumsum | True **False** |
| Euclidean dist. $k$-NN | $agg_{token}$ | First Last **Mean** |
| | Feat | Encoded **Pre-logits** |
| | cumsum | True **False** |
| | $k$ | 1 5 **10** |
| Cosine dist. $k$-NN | $agg_{token}$ | First Last **Mean** |
| | Feat | Encoded **Pre-logits** |
| | cumsum | True **False** |
| | $k$ | 1 5 **10** |
| PCA-KMeans | $agg_{token}$ | **First** Last Mean |
| | Feat | **Encoded** Pre-logits |
| | cumsum | **True** False |
| | clusters | **16** 32 64 |
| | dim | **32** 64 128 |
| RND | $agg_{token}$ | First Last **Mean** |
| | Feat | Encoded **Pre-logits** |
| LogpZO | $agg_{token}$ | First Last **Mean** |
| | Feat | Encoded **Pre-logits** |
| Action total var. | cumsum | True **False** |
| Trans. total var. | cumsum | True **False** |
| Rot. total var. | cumsum | True **False** |
| Gripper total var. | cumsum | True **False** |
| Cluster entropy | cumsum | True **False** |
| | $\delta$ | 0.01 0.05 0.1 **0.2** 0.5 1 2 5 |
| STAC | cumsum | **True** False |
| STAC-Single | cumsum | **True** False |
| *SAFE*-LSTM | $agg_{token}$ | First Last **Mean** |
| | Feat | **Encoded** Pre-logits |
| | lr | 3e-5 1e-4 **3e-4** 1e-3 |
| | $\lambda_{reg}$ | **1e-3** 1e-2 1e-1 |
| *SAFE*-MLP | $agg_{token}$ | First **Last** Mean |
| | Feat | Encoded **Pre-logits** |
| | lr | 3e-5 **1e-4** 3e-4 1e-3 |
| | $\lambda_{reg}$ | 1e-3 **1e-2** 1e-1 |

Table 10: Grid-searched and best-performing hyperparameters (in bold text) for $\pi_0$+LIBERO (left) and $\pi_0^*$+SimplerEnv (right).

| Method | HParams | Values |
|---|---|---|
| Mahalanobis dist. | $agg_{\text{hori}}$ | **First** Last First&Last |
| | $agg_{\text{diff}}$ | First **Last** First&Last |
| | cumsum | True **False** |
| Euclidean dist. $k$-NN | $agg_{\text{hori}}$ | **First** Last First&Last |
| | $agg_{\text{diff}}$ | First **Last** First&Last |
| | cumsum | True **False** |
| | $k$ | 1 5 **10** |
| Cosine dist. $k$-NN | $agg_{\text{hori}}$ | **First** Last First&Last |
| | $agg_{\text{diff}}$ | First **Last** First&Last |
| | cumsum | True **False** |
| | $k$ | 1 5 **10** |
| PCA-KMeans | $agg_{\text{hori}}$ | **First** Last First&Last |
| | $agg_{\text{diff}}$ | **First** Last First&Last |
| | cumsum | True **False** |
| | clusters | **16** 32 64 |
| | dim | 32 64 **128** |
| RND | $agg_{\text{hori}}$ | **First** Last First&Last |
| | $agg_{\text{diff}}$ | First **Last** First&Last |
| LogpZO | $agg_{\text{hori}}$ | First Last **First&Last** |
| | $agg_{\text{diff}}$ | **First** Last First&Last |
| Action total var. | cumsum | True **False** |
| Trans. total var. | cumsum | True **False** |
| Rot. total var. | cumsum | **True** False |
| Gripper total var. | cumsum | True **False** |
| Cluster entropy | cumsum | True **False** |
| | $\delta$ | 0.01 0.05 0.1 0.2 0.5 1 2 **5** |
| STAC | cumsum | True **False** |
| STAC-Single | cumsum | True **False** |
| *SAFE*-LSTM | $agg_{\text{hori}}$ | **First** Last First&Last |
| | $agg_{\text{diff}}$ | First **Last** First&Last |
| | lr | 1e-5 3e-5 1e-4 3e-4 **1e-3** |
| | $\lambda_{reg}$ | **1e-3** 1e-2 1e-1 |
| *SAFE*-MLP | $agg_{\text{hori}}$ | **First** Last First&Last |
| | $agg_{\text{diff}}$ | First **Last** First&Last |
| | lr | 1e-5 **3e-5** 1e-4 3e-4 1e-3 |
| | $\lambda_{reg}$ | **1e-3** 1e-2 1e-1 |

| Method | HParams | Values |
|---|---|---|
| Mahalanobis dist. | $agg_{\text{hori}}$ | First Last **Mean** First&Last |
| | $agg_{\text{diff}}$ | First **Last** Mean First&Last |
| | cumsum | True **False** |
| Euclidean dist. $k$-NN | $agg_{\text{hori}}$ | First Last **Mean** First&Last |
| | $agg_{\text{diff}}$ | First Last **Mean** First&Last |
| | cumsum | True **False** |
| | $k$ | 1 **5** 10 |
| Cosine dist. $k$-NN | $agg_{\text{hori}}$ | First Last **Mean** First&Last |
| | $agg_{\text{diff}}$ | First Last **Mean** First&Last |
| | cumsum | True **False** |
| | $k$ | 1 5 **10** |
| PCA-KMeans | $agg_{\text{hori}}$ | First Last **Mean** First&Last |
| | $agg_{\text{diff}}$ | First **Last** Mean First&Last |
| | cumsum | True **False** |
| | clusters | 16 32 **64** |
| | dim | **32** 64 128 |
| RND | $agg_{\text{hori}}$ | First **Last** Mean First&Last |
| | $agg_{\text{diff}}$ | First **Last** Mean First&Last |
| LogpZO | $agg_{\text{hori}}$ | First Last **Mean** First&Last |
| | $agg_{\text{diff}}$ | First **Last** Mean First&Last |
| Action total var. | cumsum | True **False** |
| Trans. total var. | cumsum | True **False** |
| Rot. total var. | cumsum | **True** False |
| Gripper total var. | cumsum | True **False** |
| Cluster entropy | cumsum | True **False** |
| | $\delta$ | 0.01 0.05 0.1 0.2 0.5 **1** 2 5 |
| STAC | cumsum | True **False** |
| STAC-Single | cumsum | True **False** |
| *SAFE*-LSTM | $agg_{\text{hori}}$ | First Last **Mean** First&Last |
| | $agg_{\text{diff}}$ | First Last Mean **First&Last** |
| | lr | 1e-4 3e-4 **1e-3** 1 |
| | $\lambda_{reg}$ | 1e-3 **1e-2** 1e-1 1 |
| *SAFE*-MLP | $agg_{\text{hori}}$ | **First** Last Mean First&Last |
| | $agg_{\text{diff}}$ | First Last Mean **First&Last** |
| | lr | 1e-4 **3e-4** 1e-3 |
| | $\lambda_{reg}$ | **1e-3** 1e-2 1e-1 1 |

Table 11: Grid-searched and best-performing hyperparameters (in bold text) for $\pi_0$-FAST on real-world rollouts.

| Method | HParams | Values |
|---|---|---|
| Max prob. | cumsum | **True** False |
| Avg prob. | cumsum | **True** False |
| Max entropy | cumsum | **True** False |
| Avg entropy | cumsum | **True** False |
| Mahalanobis dist. | cumsum | **True** False |
| Euclidean dist. $k$-NN | cumsum | **True** False |
| | $k$ | 1 **5** 10 |
| Cosine dist. $k$-NN | cumsum | **True** False |
| | $k$ | 1 **5** 10 |
| PCA-KMeans | cumsum | **True** False |
| | clusters | 16 **32** 64 |
| | dim | 32 64 **128** |
| STAC-Single | cumsum | **True** False |
| *SAFE*-LSTM | lr | 1e-4 3e-4 **1e-3** 3e-3 |
| | $\lambda_{reg}$ | 1e-3 **1e-2** 1e-1 |
| *SAFE*-MLP | lr | 1e-4 **3e-4** 1e-3 3e-3 |
| | $\lambda_{reg}$ | **1e-3** 1e-2 1e-1 |

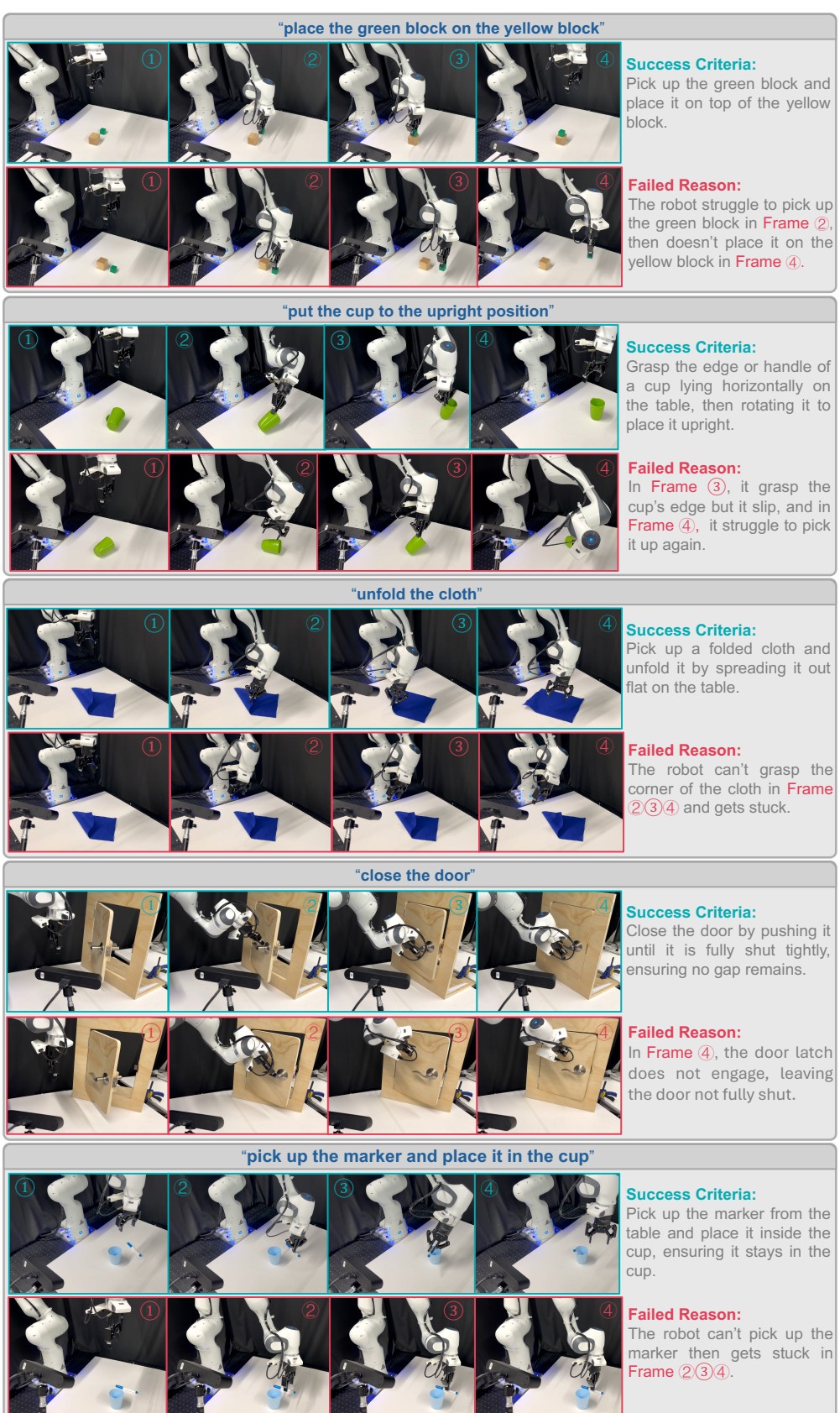

Figure 7: Example successful and failed rollouts from real-world experiments.

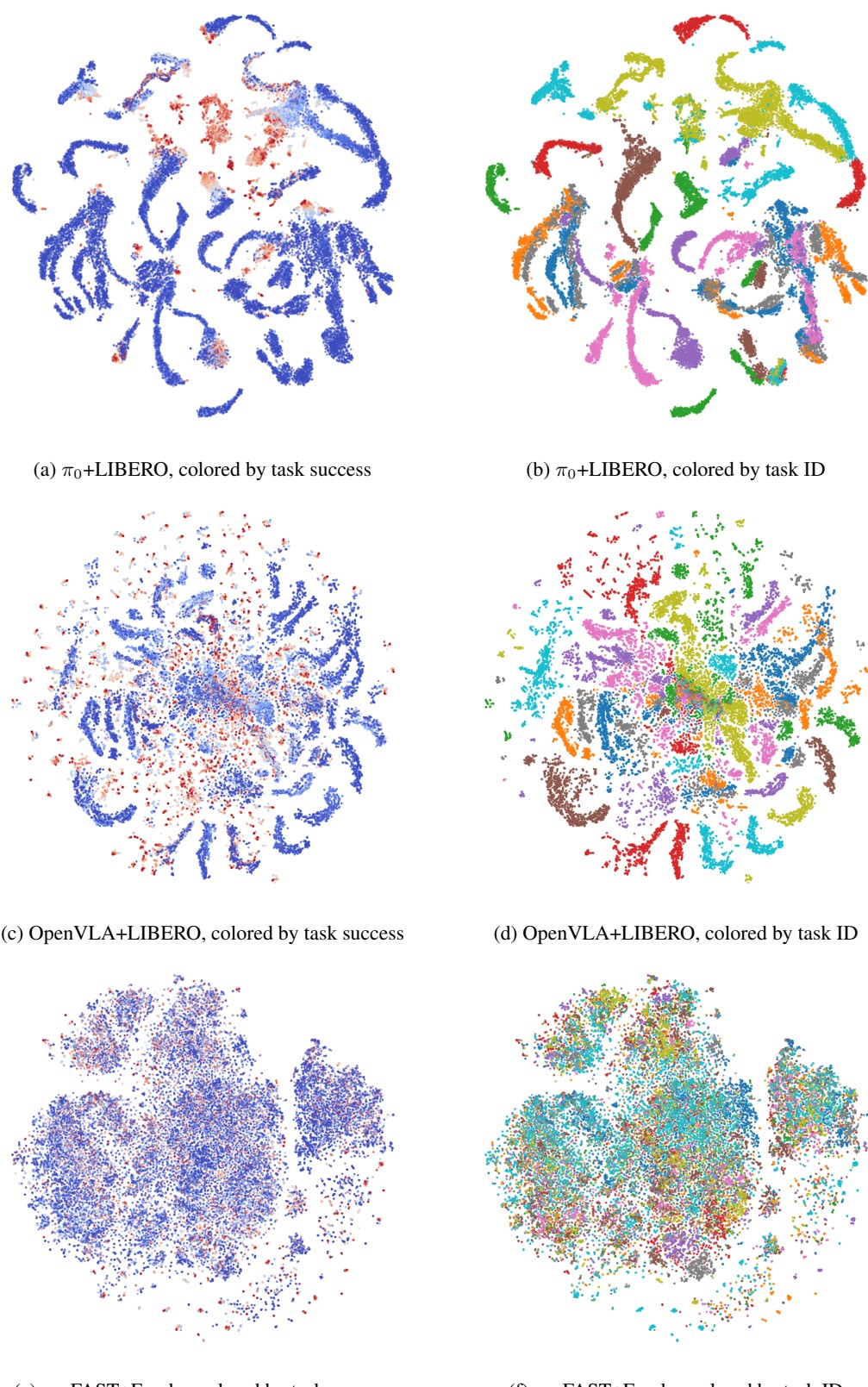

(a) $\pi_0$+LIBERO, colored by task success

(b) $\pi_0$+LIBERO, colored by task ID

(c) OpenVLA+LIBERO, colored by task success

(d) OpenVLA+LIBERO, colored by task ID

(e) $\pi_0$-FAST+Franka, colored by task success

(f) $\pi_0$-FAST+Franka, colored by task ID

Figure 8: t-SNE plots of VLA's internal features, from different evaluation benchmarks.

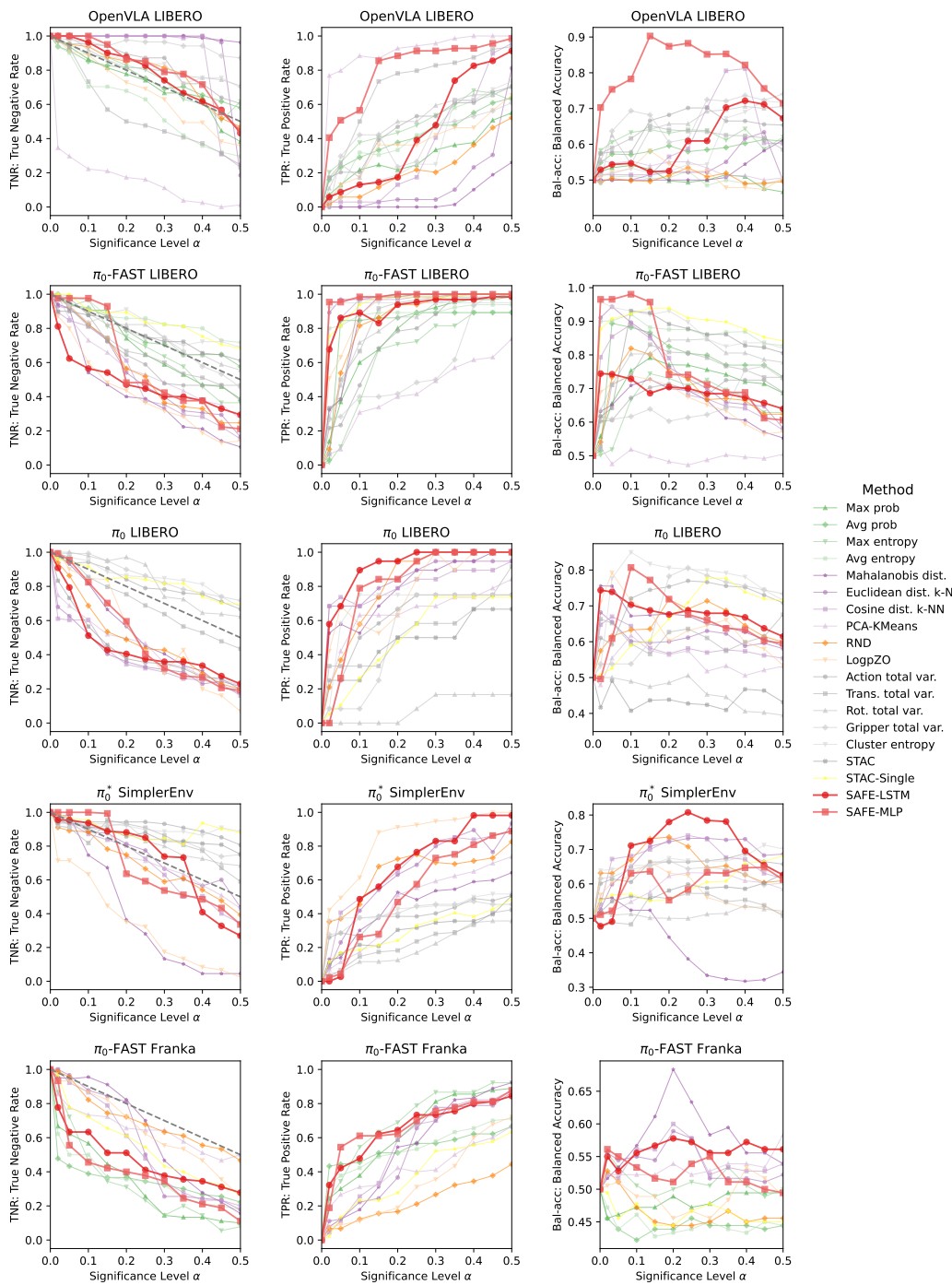

Figure 9: Additional failure detection results using $\delta_t$ obtained by functional CP. These plots show TNR (left column), TPR (middle column), and Bal-acc (right column) w.r.t. the significance level $\alpha$, for each evaluation benchmark. These plots are obtained with random seed$= 0$.

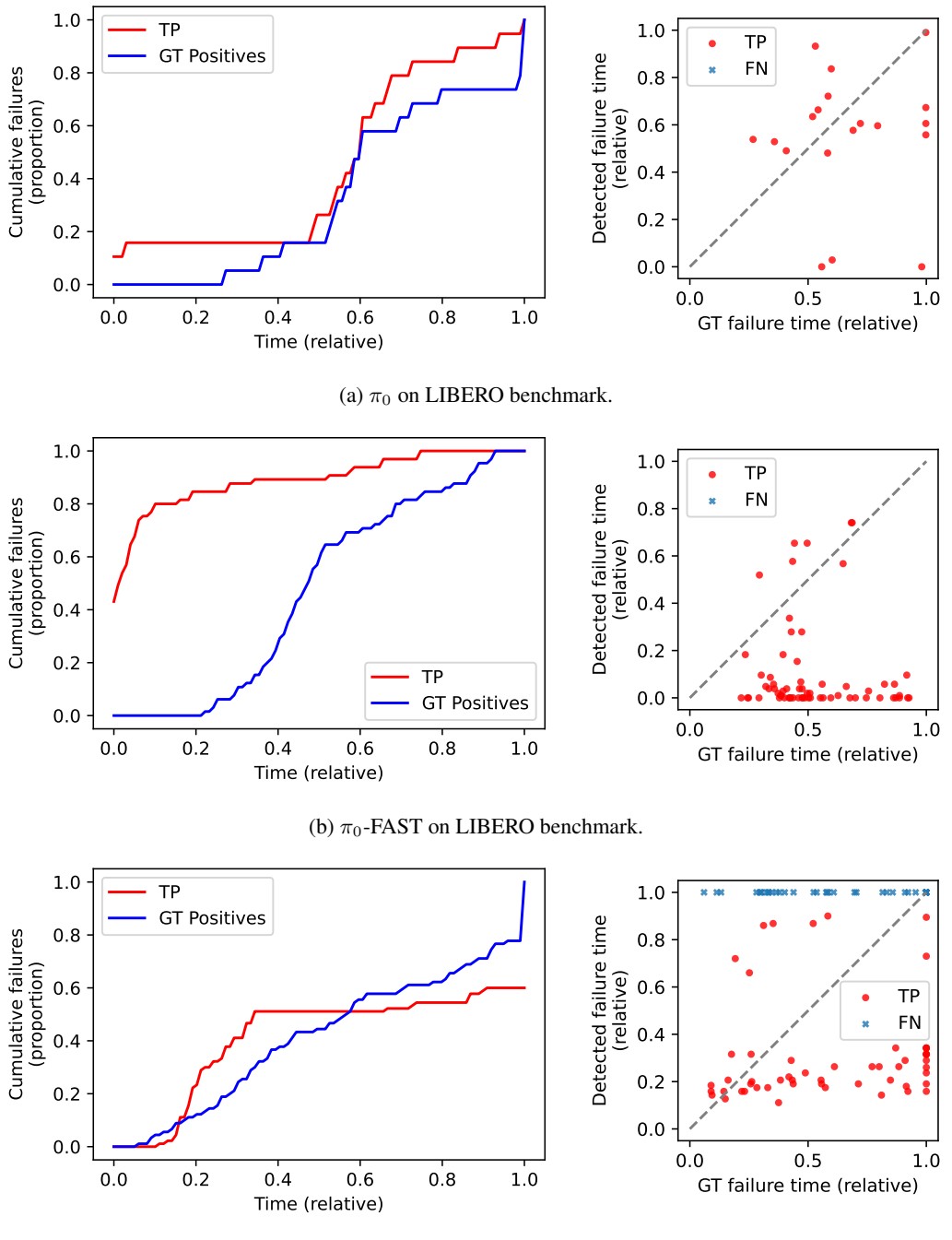

(a) $\pi_0$ on LIBERO benchmark.

(b) $\pi_0$-FAST on LIBERO benchmark.

(c) $\pi_0$-FAST on the real Franka robot.

Figure 10: Comparison between detected and ground truth (GT) failure w.r.t time. On the left column, we plot the cumulative number of true failures (true positives) detected by *SAFE*-MLP (red) and a human annotator (blue), w.r.t. elapsed time in each rollout. The right column shows the time of failures detected by *SAFE*-MLP (y-axis) and a human annotator (x-axis) for each rollout, where failures missed by the detector (false negatives) are plotted in blue crosses. Experiments are done with seed 0 and functional CP with significance level $\alpha = 0.15$.

