# OpenReview forum: "SAFE: Multitask Failure Detection for Vision-Language-Action Models"
_NeurIPS.cc/2025/Conference — NeurIPS 2025 poster_

### Official Review · Reviewer_st8r · 2025-06-30

**Clarity:** 4
**Significance:** 4
**Originality:** 3
**Rating:** 5
**Confidence:** 4

**Summary:**

This paper tackles the problem of **multitask failure detection** for large vision-language-action (VLA) robot policies deployed zero-shot on novel tasks. The authors observe via t-SNE that final-layer embeddings from successful and failed rollouts cluster into distinct regions (“failure zone”) regardless of task. Motivated by this, they propose **SAFE** (ScAlable Failure Estimation):

1. **Feature Extraction**: At each timestep, extract the VLA’s final-layer latent vector.
2. **Lightweight Probe Training**: Train a small MLP or an LSTM over features from multiple seen tasks to regress a scalar failure score st.
3. **Threshold Calibration**: Use functional conformal prediction on successful seen-task rollouts to derive a time-varying threshold δt that guarantees the false-positive rate does not exceed a user-specified α.

They evaluate SAFE on three VLA models (OpenVLA, π₀, π₀–FAST) across two simulation benchmarks (LIBERO-10, SimplerEnv) and a real Franka Panda setup. SAFE outperforms diverse baselines—token- and sample-based uncertainty, embedding-distance and OOD detectors, and action-consistency methods—achieving state-of-the-art ROC-AUC on unseen tasks and the best trade-off between detection speed and accuracy.

**Questions:**

1. Have the author try concatenating or aggregating embeddings from intermediate transformer layers, rather than only the final layer, to detect failures even earlier?

2.  How sensitive is the conformal threshold band to the choice of α in high-risk settings? Could α be adapted online based on task-specific risk profiles?

3. Can the scalar failure score be extended to a multi-dimensional output (e.g., grasp failure vs. collision) to support targeted recovery strategies?

**Ethical Concerns:**

["NO or VERY MINOR ethics concerns only"]

**Final Justification:**

I remain positive about this work, recognizing its potential contributions to the field of embodied AI. As noted, the manuscript effectively addresses the challenge of failure detection, a persistent and complex issue in this domain. Although straightforward, the method introduces a nearly cost-free solution that could significantly enhance a variety of existing VLAs. The analysis and its outcomes provide valuable insights to society. However, while the method itself holds value, addressing additional aspects such as investigation into module design, feature selection, and types of failures is also crucial. These elements are essential for rounding out the research, potentially increasing the societal impact of the work.

**Limitations:**

Yes.

**Paper Formatting Concerns:**

None.

The manuscript adheres to NeurIPS formatting guidelines: section headings, figure/table numbering, and citation style all conform properly.

**Quality:**

3

**Strengths And Weaknesses:**

**Strengths**

- Comprehensive empirical evaluation across multiple VLA models, tasks, and both simulation and real-robot settings, with multiple random seeds and error bars ensuring statistical confidence.
- Formal control of false positives via functional conformal prediction, providing theoretical guarantees.
- Well-organized presentation, clear diagrams, and consistent notation.
- Tackles a critical safety problem for zero-shot deployment of generalist VLAs in a novel multi-task setting, demonstrating substantial gains over both specialist and generic uncertainty-based baselines.
- First systematic study of multitask failure detection in VLA policies, with the insightful discovery of a task-agnostic “failure zone” in the embedding space.

**Weaknesses**

- The reliance on simple one- or two-layer probes may underutilize richer representations; unclear how performance scales if future VLAs integrate deeper multi-layer feature fusion or cross-modal attention architectures.
- Experiments focus exclusively on manipulation tasks and final-layer embeddings; the method’s robustness under radical domain shifts (e.g., navigation, aerial manipulation, or action-less video inputs) and when aggregating mid-level features remains untested.
- Core components (feature probing, conformal calibration) are adaptations of existing techniques; the primary novelty resides in their tailored application to VLA failure detection rather than introducing fundamentally new algorithms.
- Focuses solely on early failure prediction without investigating strategies for recovery or intervention once a failure is detected.

---

> ### Author Rebuttal · Authors · 2025-07-30
>
> &nbsp;
>
> We thank the reviewer for the comprehensive feedback! We are grateful that the review acknowledged that our paper is a **pioneer systematic study with comprehensive evaluation, insightful discovery, and well-organized presentation**, and also that SAFE tackles a **critical safety problem, provides formal control with theoretical guarantees, and demonstrates substantial gains over baselines**!
>
> &nbsp;
>
> # W1, Q1: “One- or two-layer probes may underutilize richer representations; unclear how performance scales if future VLAs integrate deeper multi-layer feature fusion or cross-modal attention architectures.” “Try concatenating or aggregating embeddings from intermediate transformer layers?”
>
> We thank the reviewer for bringing up this point! We maximized the simplicity and transferability when designing SAFE. By only taking feature vectors at the last layer, the proposed method can be easily integrated into any VLA models with minimal implementation changes and no finetuning on the VLAs themselves.
>
> However, we do agree that fusing or aggregating deep features from multiple layers is a promising direction and can be beneficial for failure detection. Related works have shown potential in this direction. For example, [Ref1] proposed Truthfulness Separator Vector (TSV), which is injected in the LLM latent features in the middle of the transformer and is optimized to better separate the hallucinated and truthful responses in the final token feature space. We think a similar technique can also be developed for VLA failure detection. However, this would require a special design and implementation for each VLA model (as some VLA models output discrete tokens, and others use flow matching to output continuous actions, there may not be a single design that can be applied to all VLA architectures), reducing its transferability. We leave the development of such methods as a promising future work.
>
> We also agree that simply using the latent feature from an intermediate transformer block may lead to better failure detection performance than the very last layer. As shown by [49] and [53], LLM latent features from different layers have different performance on hallucination detection, and the best one may not be the last layer. However, exactly which layer works the best depends on the model and requires extensive ablation experiments to find out. For example, as reported by [49], for the OPT-6.7B model, the 20th layer works the best, but for LLAMA2-7B, the 16th layer works the best. To locate the best layer, [49] has to perform a grid search over each LLM tested. On the contrary, in our setting, VLA users can avoid such grid search experiments and simply choose the final layer for failure detection. Therefore, we think that precisely finding the layer that works the best for VLA failure detection is outside the scope of this paper, but it would be very interesting to explore for future work.
>
> [Ref1] Steer LLM Latents for Hallucination Detection, ICML 2025
>
>
> # W2: “The method’s robustness under radical domain shifts (e.g., navigation, aerial manipulation, or action-less video inputs)”
>
> We thank the reviewer for mentioning this. SAFE only requires access to the latent features from VLA models and is broadly applicable to other domains as long as they use a white-box robot policy based on neural networks. In this work, we focus on robot manipulation tasks and VLA policies because dexterous manipulation is widely accepted to be one of the most challenging domains for VLAs, where diverse failure modes exist. We leave the application and the evaluation to other robot and non-robot (action-less video) domains for future work.
>
>
> # W3: “Adaptations of existing techniques; the primary novelty resides in their tailored application to VLA failure detection rather than introducing fundamentally new algorithms.”
>
> We agree with the reviewer that SAFE is indeed adapted from existing techniques. However, we would also like to thank the reviewer for pointing out contributions in multiple aspects besides technical novelty, including (1) the first systematic study of multitask failure detection in VLA policies, (2) the insightful discovery of a task-agnostic “failure zone”, (3) comprehensive empirical evaluation across multiple settings, (4) formal control of false positives via functional conformal prediction, and (5) demonstrating substantial gains over baselines in a novel multitask setting.
>
>
> # W4: “Without investigating strategies for recovery or intervention once a failure is detected.”
>
> Following prior works [8,17], we also primarily focus on detecting failures accurately and in a timely manner. As long as the failure is detected, the robot can either abort potentially dangerous actions or a human monitor can step in and take over control.
>
> Designing a fallback policy or an automatic failure recovery algorithm is a challenging and open research problem. We think this is beyond the scope of this work, and is a promising future direction.
>
>
> # Q2.1: "How sensitive is the conformal threshold band to the choice of α in high-risk settings?”
>
> Please refer to Figure 8 in the appendix for a comprehensive list of how TPR, TNR, and balanced accuracy change with $\alpha$. From Figure 8, we can observe that while the metrics do vary with the $\alpha$, choosing $\alpha=0.15$ (or in general, between 0.05 and 0.2) performs well across the board. We have also chosen $\alpha$ to be 0.15 for most qualitative results and analyses reported in the paper.
>
>
> # Q2.2: “Could α be adapted online based on task-specific risk profiles?”
>
> Yes, the proposed method and baselines can be extended to online or adaptive conformal prediction (CP) [Ref1]. In such a framework, rollout results are observed one-by-one and compared to the prediction results, and the significance level $\alpha$ is adjusted if the prediction is wrong.
>
> However, when VLA policies are deployed, they may constantly meet novel environments and customized task instructions, and may rarely repeat the same task. In such a case, it can be hard or even impractical to develop an adaptive CP band for each specific task. Therefore, in this work, we focus on the performance of failure detectors when they are directly deployed on a novel task, and have never seen or repeated the task before. In this setting, offline CP is more appropriate.
>
> Nevertheless, we believe online CP is an interesting extension to our work, and we leave it as an important future work.
>
> [Ref1] Adaptive conformal inference under distribution shift. NeurIPS 2021
>
>
> # Q3: “Extended to a multidimensional output (grasp vs. collision) for targeted recovery strategies?”
>
> Yes, the proposed method can be extended to a multi-class classifier that categorizes failures into different types. However, this would require annotations on failure types and demands manual efforts. SAFE aims to detect failures with minimal annotation effort (only labels on failure vs success), and we leave the multidimensional output extension as a valuable future work.
>
> Moreover, as described in Appendix Section B.3, sometimes the timing and type of failure can be ambiguous and hard to determine. Therefore, the multi-dimensional extension will be non-trivial and require extra considerations.

---

> ### Comment · Reviewer_st8r · 2025-08-07
>
> Thanks for the detailed responses.  I remain positive about this work, recognizing its potential contributions to the field of embodied AI. As noted, the manuscript effectively addresses the challenge of failure detection, a persistent and complex issue in this domain. Although straightforward, the method introduces a nearly cost-free solution that could significantly enhance a variety of existing VLAs.  The analysis and its outcomes provide valuable insights to society. However, while the method itself holds value, addressing additional aspects such as investigation into module design, feature selection, and types of failures is also crucial. These elements are essential for rounding out the research, potentially increasing the societal impact of the work.

---

> > ### Author Response · Authors · 2025-08-07
> >
> > Thanks a lot for the comment! We are grateful for your positive review and your appreciation of the contributions of our work!
> >
> > If you are interested, please also take a look at the new experiments in our rebuttal to other reviewers. They further demonstrate the generalizability of the proposed method, despite only working with the last-layer features.
> >
> > Nevertheless, we also highly recognize the value of the potential future work directions as mentioned by the reviewer, and we will explore them in the future!

---

### Official Review · Reviewer_XCVQ · 2025-07-03

**Clarity:** 3
**Significance:** 2
**Originality:** 2
**Rating:** 4
**Confidence:** 3

**Summary:**

This paper introduces SAFE, a lightweight failure detection method for vision-language-action (VLA) models. SAFE uses internal latent features to estimate failure likelihood in real time, even before visible errors occur. It outputs a single scalar score and applies conformal prediction to set calibrated thresholds. The authors evaluate SAFE on both simulated and real-world tasks, showing that it generalizes well to unseen tasks.

**Questions:**

- How important is the diversity of failure modes and tasks in the training data for SAFE to generalize well?
- How long does it take to collect a balanced set of successes and failures? Is it necessary to use VLA rollouts? Or can one use tele-operated successful and failed demos to achieve the same?
- Have you compared with learning the score using other latent representations other than VLAs?
- How would the failures detected be used to steer/control VLA behavior?

**Ethical Concerns:**

["NO or VERY MINOR ethics concerns only"]

**Final Justification:**

The approach is technically sound and the experiments are thorough. The only concern is that this approach may not be very practical due to the number of rollouts required is very high (~1000 trajectories). This also introduces a chicken-and-egg problem in which one needs to roll out these policies to collect large quantities of failed trajectories before you can quantify safety and eventually prevent policies act in risky manner.

**Limitations:**

yes

**Quality:**

3

**Strengths And Weaknesses:**

Failure detection is an important topic for realworld robotics applications. The proposed method is generalizable across tasks, efficient at runtime, and capable of detecting failures early, even before they are externally visible as shown in the experiments. It does not require per-timestep annotations, making it simple to supervise. SAFE’s use of conformal prediction provides statistical control over false positives. It achieves strong performance in both simulation and real-world settings, using minimal supervision and without relying on heavy computation.

However, the proposed supervised method does not provide true uncertainty quantification; it learns to detect patterns distinguishing success and failures but does not estimate calibrated probabilities of success of the VLA. Detecting OOD states is fundamentally a different problem than detecting failure states conditioned on a task. Failed states in certain tasks may be successful states in other tasks (e.g. opening vs closing a door), while both tasks may present in the training data. Instead of comparing with uncertainty quantification baselines, it would be more useful to test learning failure detectors using latent representation from different pretrained models, e.g. how does Dino features work vs VLA features? Does the proposed method work across different VLA embeddings?

---

> ### Author Rebuttal · Authors · 2025-07-30
>
> &nbsp;
>
> We thank the reviewer for the insightful review! We appreciate that the reviewer points out that our method is **generalizable across tasks, efficient at runtime, and capable of detecting failures early**, and also that SAFE is **simple to supervise, provides statistical control, and achieves strong performance without needing heavy computation**!
>
> &nbsp;
>
>
> # W1: “Does not provide true uncertainty quantification … does not estimate calibrated probabilities of success... Detecting OOD states is fundamentally a different problem...”
>
> SAFE does not aim to achieve uncertainty quantification (UQ) or OOD detection, but it does model the probability of failures and can detect failures of VLA policies.
>
> These 3 techniques are closely connected but have subtle differences. Here, we provide a discussion about them.
>
> **Failure detection** is the task of detecting failures when a robot is performing certain tasks. SAFE learns the likelihood of failure through training on a set of successful and failed rollouts. SAFE-LSTM is trained by BCE loss and outputs a normalized score indicating the probability of failure of VLA. The output of SAFE-MLP is not normalized and thus not a probability. However, output scores from both SAFE-LSTM and SAFE-MLP are calibrated through functional Conformal Prediction CP and can be used for failure detection with theoretical guarantees.
>
> **Uncertainty quantification** measures a VLA’s uncertainty in its outputs and can be used as a proxy for failure detection. In our experiments, the token uncertainty baselines and sample consistency baselines are inspired by LLM/VLM literature and designed based on UQ. Methods in this category are typically training-free, but they only show limited success according to our experiments.
>
> **OOD detection**-based failure detection methods treat successful rollouts as normal execution conditions and assume that deviations from this norm lead to a higher chance of failure. In our experiments, the embedding distribution-based baselines are designed to detect policy failure based on OOD detection. Methods in this category can work without failed rollouts. In our experiments, we adapted them to take in both successes and failures, and they showed strong performance. Please see Section 2.2 in our paper and also [8] for comprehensive discussions on these methods.
>
> Uncertainty quantification (UQ) methods have been widely used for LLM/VLM hallucination detection (see section 2.3), and OOD detection-based methods have been shown to be effective for failure detection in robotics policies (see Section 2.2). Therefore, we think it’s appropriate to use them as baselines.
>
> SAFE directly learns to detect failures from a history of observations and the language instruction specifying the desired task without using uncertainty or OOD detection as the proxy measurement. Experiments show that this direct learning regime used by SAFE is more effective and achieves better performance than other methods. We will add more discussion about this in the camera-ready version.
>
> # W2: “Failed states in certain tasks may be successful states in others.”
>
> We want to clarify a potential misunderstanding around this point. Success and failure criteria are indicated in the given language instruction of the VLA policy. VLA models take in the language instruction together with observation, process them through a deep neural network, and aggregate the information in a high-level feature vector. These feature vectors have extracted the success criteria from the language input and can identify failures based on both the observations and the language instructions specifying the desired task. Our analysis and experiments show that SAFE can detect failures across different tasks, including unseen ones, even when the same states are visited.
>
>
> # W3, Q3: “Using latent representation from different pretrained models, e.g. Dino?”
>
> In the following table, we ablate SAFE-MLP and SAFE-LSTM using DINOv2 features, CLIP features, DINOv2 and CLIP concatenated (DINOv2+CLIP), and the VLA last-layer features (VLA; our main method). DINOv2 and CLIP features are extracted from the observation images, and this experiment is conducted over the real-world Franka rollouts, following the same setting as reported in the paper. Numbers are ROC-AUC on the Seen and Unseen tasks.
>
> | Method | LSTM| LSTM |MLP |MLP |
> |-|-|-|-|-|
> | Eval Task Split|Seen|Unseen|Seen|Unseen|
> | DINOv2|76.93|56.96|76.20|59.46|
> | CLIP|76.77|52.71|77.88|59.77|
> | DINOv2+CLIP|77.09|59.65|76.36|58.43|
> | VLA|77.27|58.70|86.76|64.16|
>
> The best performing method in the above table is the SAFE-MLP method based on VLA last-layer features, where VLA features outperform other feature types by a large margin. We think that this is because VLA feature space learns high-level information about the tasks, and thus can more easily distinguish failures from successes than general pretrained models. Similar findings were also reported in related works like [17].
>
> # Q1: “How important is the diversity of failure modes and tasks in training?”
>
> We thank the reviewer for bringing this up. Yes, since SAFE learns to distinguish failures from successes from training rollouts, it is important to have good coverage of the diversity of failure modes and tasks in training. To quantify this effect, we conduct an experiment varying the number of seen tasks that are used in training. Note that different tasks typically also have different failure modes, and in this way, we are also ablating the diversity of failure modes.
>
> In the following table, we report the failure detection ROC-AUC on the OpenVLA+LIBERO benchmark, trained on different numbers of tasks. While the number of seen tasks is ablated, all experiments use the same set of unseen tasks for evaluation, and performance on unseen tasks is comparable. All numbers are averaged over 3 random seeds. Experiments with 7 tasks for training match the setting reported in the paper. Training-free methods do not depend on training tasks and are not shown.
>
> | # Training Tasks|1|1|3|3|5|5|7|7|
> |-|-|-|-|-|-|-|-|-|
> | Eval Task Split|Seen|Unseen|Seen|Unseen|Seen|Unseen|Seen|Unseen|
> | Mahalanobis|40.21| 52.75| 58.00| 52.31| 57.68| 50.78| 62.03| 58.85|
> | Euclid.kNN|49.74| 63.76| 61.66| 67.02| 59.14| 67.11| 66.00| 55.23|
> | Cosine.kNN|53.27| 60.76| 65.39| 65.64| 67.46| 70.57| 67.09| 69.45|
> | PCA-KMeans|60.39| 40.58| 61.18| 52.87| 61.50| 53.06| 57.18| 55.10|
> | RND|29.29| 50.32| 54.46| 47.39| 56.71| 49.15| 52.57| 46.88|
> | LogpZO|61.75| 56.17| 52.89| 50.49| 65.99| 56.60| 61.57| 52.91|
> | SAFE-LSTM|50.88| 52.25| 68.85| 63.31| 70.70| 66.31| 70.24| 72.47|
> | SAFE-MLP|54.34| 63.76| 67.86| 67.03| 69.32| 68.17| 72.68| 73.47|
>
> The above table shows that for most methods, with more tasks used for training, the performance on unseen tasks gets better. The proposed SAFE-MLP performs well in all settings and can also achieve good performance when fewer (3 or 5) tasks are used for training.
>
> # Q2-1: “How long does it take to collect a balanced set of successes and failures?”
>
> The time used to collect a balanced set of real-world rollouts depends on the performance of the VLA model. During our data collection, we adjusted the environment setup and task difficulty with the target of achieving such a balance.
>
> In total, by deploying the pretrained pi0-FAST model on a Franka robot, we collected 1307 rollouts with 420 successes and 887 failures, among them 390 successful and 390 failed ones are used for training and evaluation. Each rollout typically takes 40~60 seconds to collect, and the total collection time for the real-robot rollouts takes roughly 10 days.
>
> # Q2-2: “Necessary to use VLA rollouts? Can one use teleoperated demos?”
>
> Yes, it’s possible to use teleoperated demos to train the failure detector. SAFE only requires the VLA’s internal features, which can be obtained by running inference over rollouts collected by human demonstration or another policy.
>
> However, such “off-policy” rollout collection may contain different failure modes from VLA’s (i.e., VLA policies and human operators may fail in different ways), and therefore, the failure detectors trained on teleoperated rollouts may have worse performance when directly applied to the VLA (“on-policy”) rollouts.
>
> # Q4: “How would the failures detected be used to steer/control VLA behavior?”
>
> Our work focuses on detecting failures accurately and in a timely manner, which enables either the robot to abort potentially dangerous actions or a human monitor to step in and take over control. How to learn a recovery policy or how to improve the VLAs themselves are important areas for future work. In this work, we focused on detecting failures of multitask VLAs reliably and in real-time, which is a crucial stepping stone towards autonomous recovery (e.g., with a fallback policy) and policy improvement (e.g., through interactive imitation learning).
>
> We think it is possible to use the findings from this paper to further develop methods for steering or improving VLA behaviors. For example, we show that the embeddings for successful and failed rollouts are separated in the latent space, so it is possible to learn a steering vector that manipulates the latent activations of a VLA and changes its output actions, as done in [Ref1] and [Ref2]. However, different from the stylization or hallucination reduction tasks for LLMs, robot manipulation involves multistep closed-loop interaction between the policy and the environment, which greatly complicates the relationship between VLA outputs and task execution successes. Therefore, how to improve VLAs through activation steering is a challenging and open research question beyond the scope of our paper.
>
> [Ref1] Inference-time intervention: Eliciting truthful answers from a language model. NeurIPS 2023
>
> [Ref2] Adaptive activation steering: A tuning-free llm truthfulness improvement method for diverse hallucinations categories. ACM TheWebConf 2025

---

> > ### Comment · Reviewer_XCVQ · 2025-08-06
> > **Thanks for the response**
> >
> > Thanks for the detailed response and providing additional results. My questions have been mostly addressed and I am happy to raise my rating to 4.
> >
> > The only concern I have is that the number of rollouts required is surprisingly high (~1000 trajectories) and this also introduces a chicken-and-egg problem in which one needs to roll out these policies to collect large quantities of failed trajectories before you can quantify safety and eventually prevent these policies act in risky manner. This makes this approach unappealing in practice.

---

> ### Author Response · Authors · 2025-08-06
>
> We appreciate the reviewer for acknowledging our response and adjusting the score!
>
> To clarify, SAFE only needs around 200 - 400 rollouts for training, and a large portion of collected rollouts are used for evaluation but not training. In the following table, we provide the detailed statistics on the rollouts collected for each benchmark and how they are split into training and evaluation subsets.
>
> | Benchmark | Total Tasks | Train Rollouts | Eval Seen Rollouts | Eval Unseen Rollouts | Total Rollouts |
> |-|-|-|-|-|-|
> | LIBERO | 10 | 210 | 140 | 150 | 500 |
> | SimplerEnv - Google Robot | 4 | 198 | 102 | 100 | 400 |
> | SimplerEnv - WidowX  | 4 | 198 | 102 | 100 | 400 |
> | Real Franka | 13 | 450 | 150 | 180 | 780 |
> | Real WidowX | 8 | 250 | 133 | 149 | 532 |
>
> As SAFE is designed for multitask failure detection, it is **trained on only a limited set of training tasks and rollouts** and **can generalize to new tasks without further collecting rollouts**. While SAFE does require hundreds of rollouts from multiple tasks during training, when handling new tasks, SAFE becomes more efficient than existing task-specific failure detectors (like [8,17]) that require collecting rollouts for training and calibration for every new task encountered.
>
> Also note that in our setting, rollouts are collected by simply running VLA inference. This is a much easier task than collecting demonstration rollouts for training a robot policy, where the latter requires a human to carefully perform the task manually by demonstration or teleoperation. But in our setting, the human only needs to reset the environment, hit a button to make the VLA run and then label the rollout as success or failure.

---

### Official Review · Reviewer_bqvs · 2025-07-03

**Clarity:** 3
**Significance:** 2
**Originality:** 2
**Rating:** 4
**Confidence:** 2

**Summary:**

The paper introduces SAFE, a multitask failure detector for generalist VLA policies, which leverages internal policy features to predict failures across diverse tasks and environments; SAFE demonstrates good performance and strong generalization on both simulated and real-world benchmarks.

**Questions:**

1. How does SAFE generalize to other VLAs beyond pi0?
2. Can you provide more qualitative examples of the failure modes in real-world settings?
3. What are the real-time computational requirements for SAFE on real-world settings?

**Ethical Concerns:**

["NO or VERY MINOR ethics concerns only"]

**Final Justification:**

The rebuttal adds new results and clarifications that address most of my concerns, particularly w.r.t. broader real-world validation, additional architectures, and computational overhead. Minor questions remain about deployment in diverse settings, but the main weaknesses have been resolved.

**Limitations:**

Yes.

**Paper Formatting Concerns:**

None.

**Quality:**

2

**Strengths And Weaknesses:**

Strengths:

+ This paper proposes a unified, efficient failure detector for VLA models that generalizes to unseen tasks without task-specific tuning.

+ The authors have demonstrated strong empirical results and showed the proposed models can detect failure during early stages on a range of VLA architectures and real-world experiments.

Weakness:

+ This paper primarily evaluates the failure detector on two simulation environments, while the real-world experiments only evaluate the p0 checkpoint on the Franka Emika Panda Robot, which is quite limited.

+ SAFE only leverages the last-layer features for error detection, which might not be a meta representation space for other diverse VLA architectures such as octo; The generalizability of the proposed technique to other VLAs is unknown.

+ The authors provide limited discussion of practical deployment constraints, such as computational overhead or integration into existing robot stacks.

---

> ### Author Rebuttal · Authors · 2025-07-30
>
> &nbsp;
>
> We thank the reviewer for a careful and constructive review! We appreciate that the reviewer found our method **unified, efficient, and generalizable to unseen tasks**, and acknowledged that SAFE shows **strong empirical results and can detect failures during early stages**!
>
> &nbsp;
>
> # W1: “Evaluates the failure detector on two simulation environments, while the real-world experiments only evaluate the p0 checkpoint on the Franka Emika Panda Robot, which is quite limited.”
>
> We provide an additional set of real-world experiments as follows, where the OpenVLA model trained on the “Open-X Magic Soup++" dataset is deployed on a real WidowX robot arm in our lab. In this experiment, we collected a total of 532 rollouts on the following 8 tasks, with 244 successes and 288 failures. Each task has roughly the same number of rollouts.
>
> - Lift AAA Battery
> - Lift Eggplant,
> - Lift Red Bottle
> - Lift Blue Cup
> - Put Blue Cup on Plate
> - Put the Red Bottle into Pot
> - Put the Carrot on Plate
> - Put the Red Block into the Pot
>
> The failure detection ROC-AUC of different methods is reported in the following table. This experiment follows the same evaluation protocol as the OpenVLA+LIBERO experiment reported in the paper. All numbers are averaged over 3 random seeds. 6 of the 8 tasks are in the Seen subset, and the remaining 2 are in the Unseen subset.
>
> | Method | Seen | Unseen |
> |-|-|-|
> | Max prob. | 50.77 | 54.25 |
> | Avg prob. | 48.94 | 44.36 |
> | Max entropy | 51.88 | 49.19 |
> | Avg entropy | 47.72 | 53.84 |
> | Mahalanobis dist. | 82.37 | 70.00 |
> | Euclid. dist. kNN | 72.01 | 53.64 |
> | Cosine dist. kNN | 74.76 | 65.88 |
> | PCA-KMeans | 75.62 | 47.22 |
> | RND | 66.68 | 47.67 |
> | LogpZO | 62.94 | 51.32 |
> | SAFE-LSTM | 84.29 | 71.80 |
> | SAFE-MLP | 89.11 | 88.42 |
>
> This table shows that SAFE-MLP achieves the best failure detection results on both seen and unseen task splits on the real-world OpenVLA+WidowX setup, with a significant margin over baselines. This complements the real-world experiments we had on the Franka robot and shows the generalizability of the proposed SAFE method.
>
>
> # W2, Q1: “The last-layer features might not be a meta representation space for other diverse VLA architectures such as octo; The generalizability of the proposed technique to other VLAs is unknown.” “How does SAFE generalize to other VLAs beyond pi0?”
>
> We note that we have already tested SAFE and baselines on 3 VLA architectures, OpenVLA, pi0, and pi0-FAST. These 3 VLAs use different backbone VLMs (LLAMA2+DINOv2+SigLIP, and PaliGemma), tokenization schemes (per-dimension binning, flow matching, and frequency-space tokenization), and evaluation benchmarks (LIBERO, SimplerEnv, and real-world Franka). We believe these experiments have demonstrated the generalizability of SAFE over different VLAs.
>
> We are grateful for the reviewer’s suggestion and, in the interest of completeness, we have also included experimental results on the Octo VLA on the SimplerEnv benchmark. In this experiment, we use Octo-small as we found that Octo-small has a higher success rate (Avg success rate = 15.3%) than Octo-base (Avg success rate = 12.7%) on SimplerEnv. We would like to note that Octo is not among the most performant VLAs available in the literature: probably because the Octo model is relatively small (27M or 93M parameters) and did not use VLM as initialization, its performance is significantly lower than $\pi_0^*$ (Avg success rate 71.6%) on this benchmark.
>
> Out of 23 tasks provided in SimplerEnv, Octo-small has a 0% success rate on 7 of them, and we only use 12 tasks that have success rates > 5% for this experiment. 9 out of 12 tasks are seen, and the other 3 are unseen. Each task has 100 rollouts. The following numbers are ROC-AUC averaged over 3 random seeds, as done for other experiments in the paper.
>
> | Method | seen | unseen |
> |-|-|-|
> | Mahalanobis dist. | 92.08 | 88.92 |
> | Euclidean dist. k-NN | 89.00 | 87.80 |
> | Cosine dist. k-NN | 89.90 | 87.48 |
> | PCA-KMeans | 73.12 | 75.07 |
> | RND | 86.23 | 78.49 |
> | LogpZO | 88.46 | 71.12 |
> | Action total var. | 54.41 | 60.16 |
> | Trans. total var. | 49.85 | 61.27 |
> | Rot. total var. | 52.19 | 60.80 |
> | Gripper total var. | 54.06 | 59.04 |
> | Cluster Entropy | 50.00 | 50.00 |
> | STAC | 47.69 | 38.50 |
> | STAC-Single | 54.81 | 62.96 |
> | SAFE-MLP (Ours) | 89.93 | 83.44 |
> | SAFE-LSTM (Ours) | 90.15 | 85.29 |
>
> From the above table, we can see that Octo embedding space also shows a great potential for failure detection, as the embedding distribution baselines and SAFE all achieve high performance. Although SAFE-LSTM does not achieve the very top performance, it’s very close to the best method, with a gap of only 2 to 3 percent.
>
>
> # W3, Q3: “Practical deployment constraints, such as computational overhead or integration into existing robot stacks.”
>
> **Computational overhead**: SAFE uses a 1-2 layer MLP or LSTM and poses a minimal (<1%) computational overhead at runtime. For example, SAFE-LSTM has 2.3 million parameters and an inference time of 0.73 ms. This is negligible compared to large VLA models. For instance, pi0 has 3.3 billion parameters and an inference time of 149 ms. We will include this information in the updated paper.
>
> **Integration requirements and deployment constraints**: SAFE only requires access to the latent features of VLA models and is compatible with any white-box robot policies based on neural networks. However, SAFE does require deploying the policy and collecting successful and failed rollouts to train the failure detector before it can detect failures.
>
>
> # Q2: More qualitative examples of the failure modes in real-world settings.
>
> Unfortunately, NeurIPS does not allow us to provide PDFs, videos, or external links during rebuttal. We will include more qualitative examples for the camera-ready version.
>
> Instead, we went through all the failure rollouts collected from our real-world experiments, and provided a textual summarization of failure reasons as follows (sorted by frequency):
>
> - Grasping failure
> - Placing failure
> - Frozen action (where the predicted action becomes 0 and the robot stops moving.)
> - Unstable action (where the robot seems to be moving randomly without executing the task.)
> - Time up
> - Press failure (The robot fails to press a button.)
> - Collision with other objects / disrupting the environment.
> - Go for the wrong object (The robot approaches or picks up the wrong object.)
> - Push failure (The robot fails to push an opening drawer or an opening door.)

---

> > ### Comment · Reviewer_bqvs · 2025-08-05
> >
> > Thank you for the detailed response and additional results, which addressed most of my concerns. I have raised my score accordingly.

---

> > > ### Author Response · Authors · 2025-08-06
> > >
> > > We sincerely thank the reviewer for acknowledging our response and raising the score!
> > >
> > > Let us know if there are any remaining concerns that were not addressed. We are more than happy to provide more responses!

---

### Decision · Program_Chairs · 2025-09-17

**Decision:**

Accept (poster)

**Comment:**

This paper introduces SAFE, a lightweight and generalizable failure detection framework for VLA policies. The work tackles an important and underexplored safety challenge. The work also has strong empirical results across multiple VLAs, simulation benchmarks, and real-world robot deployments. There are some remaining concerns about the limited scope of real-world validation, reliance on final-layer features, the large number of rollouts needed for training.